# CAPTION SUPERVISION ENABLES ROBUST LEARNERS

## ABSTRACT

Vision language (VL) models like CLIP are robust to natural distribution shifts, in part because CLIP learns on unstructured data using a technique called caption supervision; the model inteprets image-linked texts as ground-truth labels. In a carefully controlled comparison study, we show that caption-supervised CNNs trained on a standard cross-entropy loss (with image labels assigned by scanning captions for class names) can exhibit greater distributional robustness than VL models trained on the same data. To facilitate future experiments with high-accuracy caption-supervised models, we introduce CaptionNet, which includes a class-balanced, fully supervised dataset with over 50,000 new human-labeled ImageNet-compliant samples which includes web-scraped captions. In a series of experiments on CaptionNet, we show how the choice of loss function, data filtration and supervision strategy enable robust computer vision. We also provide the codebase necessary to reproduce our experiments at VL Hub.

## 1 INTRODUCTION

**Motivation.** Real world uses of deep learning require predictable model behavior under data distribution shift. Our paper deals with **distributional robustness**, the effect of distribution shift on image classifiers. For brevity's sake, we may simply refer to this as **robustness**. Since 2019, numerous works have studied effective robustness on ILSVRC-2012 ImageNet classification, quantifying the impact of various interventions (Taori et al., 2020; Miller et al., 2021; Fang et al., 2022; Radford et al., 2021).

The majority of standard deep network models have been found to perform significantly worse under so-called **natural distribution shifts**, such as changes in lighting or stylized renderings of object classes (Hendrycks & Dietterich, 2019; Miller et al., 2021), leading many researchers to reconsider how close computer vision models had truly come to human-level accuracy, and underscoring the need for more robust models and diverse evaluation datasets. The now-popular ViT-L CLIP model by Radford et al. (2021) was the first vision language (VL) model to show natural distributional robustness comparable to humans across a wide range of ImageNet shifts, at the cost of some base-task accuracy. Subsequent work from Jia et al. (2021); Pham et al. (2021) showed that human-level distributional robustness is possible even as base accuracy approaches SOTA, as long as sufficient data is available for training. The gains are not limited to CLIP; other VL-loss functions also achieve strong distributional robustness (Yu et al., 2022). In our paper, we focus on CLIP, since it is publicly available and by far the most popular VL model. Since CLIP's design differs from typical models in several important ways (loss function, training dataset, the use of natural language captions as labels), it is of great interest to isolate the effect of these factors on model distributional robustness.

Recent works have addressed this question, and have reached various interesting conclusions. Fang et al. (2022) posit that the intrinsic diversity of training image data is the main cause for the distributional robustness gains of VL models *in the zero-shot setting*, with other factors such as *language supervision contributing little to no distributional robustness*. On the other hand, Santurkar et al. (2022) seemingly provide a counterpoint; given a sufficiently large pretraining dataset and *descriptive, low variability captions*, contrastively trained VL models, AKA caption-supervised models, outperform models trained with the SIMCLR-loss *in the transfer learning setting*.

Does caption supervision lead to models which perform better under distribution shift, or does it not? This question is difficult to answer conclusively, not only because of the aforementioned confounds, but also the often vast discrepancies in *base accuracy* (Taori et al., 2020). VL-loss models are

Table 1: *This table lists works which relate to ours own, evaluating distributional robustness in computer vision. We catalog the contributions of each paper with with respect to certain key factors. VL vs CE-loss indicates whether the paper conducted controlled comparisons on the effects of VL-loss (InfoNCE) and CE-loss. Captioning strategy is whether the study evaluated the effects of captioning strategy on model performance. Our paper is the first to compare CE-loss and VL-loss models trained and evaluated on multiple datasets at both low and high accuracies.*

| Citation | VL-vs-CE-loss | Non-ImageNet Training | Transfer Learning | Captioning Strategy | High Acc. Models |
|---|---|---|---|---|---|
| Recht et al. (2019) | No | No | No | No | Yes |
| Taori et al. (2020) | No | Yes | No | No | Yes |
| Radford et al. (2021) | Yes | No | Yes | No | Yes |
| Miller et al. (2021) | No | Yes | No | No | Yes |
| Fang et al. (2022) | Yes | Yes | No | Yes | No |
| Santurkar et al. (2022) | No | Yes | Yes | Yes | Yes |
| Nguyen et al. (2022) | No | Yes | No | No | No |
| **This paper** | **Yes** | **Yes** | **No** | **Yes** | **Yes** |

typically trained on massive uncurated datasets from the web, and perform poorly when restricted to training on standard benchmark datasets; collecting massive labeled datasets for CE models is an expensive undertaking. Therefore, it is difficult to conduct comparisons isolating the effects of dataset size, dataset composition, loss function, filtration method, and data supervision method.

**Our contributions.** This paper addresses the aforementioned challenges in several important ways:

1. Following the lead of Miller et al. (2021), some recent works (cf. Table 1) have extrapolated trends detected in low accuracy models to high accuracy regimes for which no models exist. Our experiments indicate that when the loss function is changed, *the validity of these extrapolations may not apply*, even when the dataset is the same. Given that several controlled studies are performed in the low accuracy regime, their results must be interpreted with appropriate caveats in mind.
2. In order to enable high-accuracy comparisons between models, we introduce CaptionNet, a 100-class image classification dataset composed of images, human-authored captions and human-annotated labels. We source these from four existing benchmark datasets, and augment with over 50,000 newly supervised creative commons samples sourced from Flickr.
3. We train cross-entropy (CE-loss) and VL-loss models models on CaptionNet, and find we are able to achieve *high base accuracy* with both architectures, allowing us to better isolate the effect of labelling strategy, as well as loss functions and architectures.
4. From our observations, we are able to conclude that the interaction between loss function and data filtration strategy contribute much more to distributional robustness than has previously been shown.
5. We also provide new insight on the impact of caption style in vision-language models, showing that improved caption supervision can have a positive impact on distributional robustness.

**Related work.** Nguyen et al. (2022) were an important precursor to the work we present here; their extensive experiments on different caption-supervised datasets in the low accuracy regime made it evident that controlling for the pretraining dataset was absolutely essential for understanding distributional robustness. We extend this result and show that the loss function and labeling strategy can also play a pivotal role in distributional robustness, even when the size of the dataset and the source of image data are the same. Unlike Nguyen et al. (2022), our results are also presented in a high-accuracy regime.

In addition to their work showing the importance of data to distributional robustness, Fang et al. (2022) introduced ImageNet-Captions, which added Flickr-captions to nearly 450,000 ImageNet images. We made use of this dataset when building CaptionNet, but added over 50,000 new human-supervised samples in order to rebalance the classes, as it has been shown that CE-loss models often struggle with class imbalances (Phan & Yamamoto, 2020)

Santurkar et al. (2022) focused on two contrastive learning models in a fixed-feature setting, where model weights are frozen and a linear probe is trained using task data. Our work compares the zero-shot performance of a generalist VL model to a CE model that is, in some sense, fully trained, but using an inherently noisy labeling technique. The choice of a different learning paradigm and a different set of controls for the model allows for a novel set of insights. We also provide new insights into the effects of captioning and filtering strategies on model distributional robustness.

## 2 SETUP AND INITIAL OBSERVATIONS

**Training Datasets and Distribution Shifts.** Our principal tool for measuring distributional robustness in this paper is accuracy on distribution shift datasets. A distribution shift is test-time data which differs in some patterned way from the data on which the model was trained. In this paper, we focus exclusively on natural distribution shifts, since this is the area in which CLIP outperformed prior methods. A natural distribution shift is a shift in the data distribution generated by some naturally occurring parameter; sunlight, changes in perspective, abstract representations of objects. We focus on the following distribution shifts: **Imagenet-Sketch (in100-s, in-s)**, **Imagenet-R (in100-r, in-r)**, **Imagenet-A (in100-a, in-a)**, and **Imagenet-V2 (in100-v2, in-v2)**.

Additional details on our pretraining datasets and distribution shifts are in the Appendix (Section A and Section B.)

**Terminology and abbreviations.** As we conducted a wide range of experiments, for convenience, we provide a list of all of the terminology and abbreviations used in our experiments throughout the paper.

*Loss functions:* This paper compares two main loss functions; **VL-loss** refers to the InfoNCE loss used by CLIP Radford et al. (2021). **CE-loss** is the typical cross-entropy + softmax classification scheme used to train the vast majority of computer vision models.

*Label types:* In this paper, the term **label** refers to whatever data source the loss function uses to train the model. CE-loss models use integer labels, and VL-loss models use caption labels. Human-selected labels we refer to as **ground-truth**. The labels generated by an algorithmic process are referred to as either **synthetic** or **subset-matched**.

*Labeling strategies:* For unsupervised datasets such as LAION and YFCC, ground-truth labels do not exist. Therefore, we must choose a labeling strategy by which the model associates labels with classes. VL-loss uses learned embeddings from natural language captions to associate text tokens and image features. But CE-loss models cannot directly learn all possible token combinations as unique classes; the set would be far too large. Instead, we employ a strategy we call **subset matching**, an extension of the "substring matching" used by Fang et al. (2022).

This strategy, illustrated in detail in Fig. 1, labels samples as follows; if a sample caption contains a matching term, then the corresponding class label is applied. If the sample caption contains no matching terms for any class, then no label is applied (the sample is filtered out of the dataset; hence we match only a subset of the dataset).

*Why subset matching?* Because we aim to conduct a controlled comparison between loss functions, we wanted to choose a labeling strategy which was model-free, consistent, and whose error was approximately normally distributed; we experiment in Table 5 with a model generated by adding random gaussian label noise to ground truth labels, and find that its E.R.R. is very similar to the subset matched models in the same table.

*Captioning strategies;* We explore multiple captioning techniques in this paper. **Flickr captions** refer to captions scraped from a combination of Flickr titles, tags and descriptions. Flickr captions are generated using either the **title** of the image, the **tags** associated with the image, the prose **description (descr)** of the image, or some combination. **ttd** captions are the union of title, tags and description; this was our default captioning strategy for Flickr-style captions. **Alt-text captions** are captions taken from alt-text image descriptions on websites. **BLIP captions** refer to captions generated using the BLIP model from Li et al. (2022). Finally, **annotated captions** refer to human authored captions which describe an image.

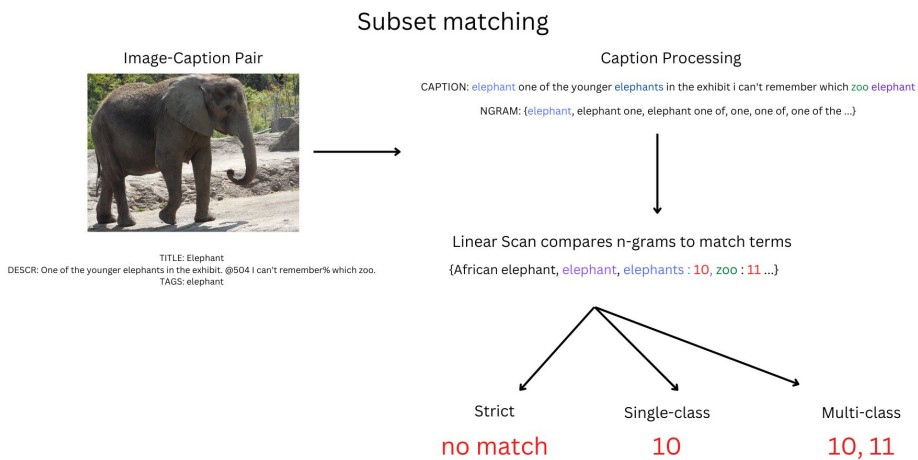

Figure 1: **Subset matching; an overview.** *Subset matching is a simple labeling strategy for unsupervised image-caption pairs. The caption is processed and converted to n-grams which are then matched against a database of terms which point to integer-label classes.*

*Matching strategies;* Subset matching requires a matching strategy, which amounts to a method for handling multiple labels on a single image. The basic idea of the three strategies, **strict (strict), single-class (sc), multiclass (mc)**, can be gleaned from Fig. 1; for additional details, please see Section E.

*Matching terms;* Subset matching depends on a database of matching terms associated with each integer label. **Ours (ours)**, **openai (oai)** and **default (def)** are the three distinct sets of labels we used for the matching algorithm. More details on these different matching term sets can be found in Section E

We also define a metric to compare filtration methods. **Dataset utilization (Ds. Util)** of a model on a dataset is the ratio of correctly labeled samples to correct + incorrect + unlabeled samples.

**Measures of effective robustness.** There is no single ideal measure of effective robustness; but taken together, we can use multiple measures to get a complete picture. One common metric, and the one we use primarily, is **average robustness (Avg. Rob.)**, the simple mean of our distribution shifts. Although this metric is easy to interpret, it does tend to gloss over the often substantial differences between shifts; therefore, we also include **shift-specific accuracy** measures in Table D. In addition, we discuss some shift-specific findings in Section A.

Another measure we use is **effective robustness**, introduced by Taori et al. (2020), primarily in order to compare our work to the existing literature. Effective robustness is described in Taori et al. (2020) as a measure of how robust a model is on a given shift, beyond what is expected from having higher accuracy on the original test set. It is generally represented as the deviation in performance with respect to a set of baseline models on ImageNet and its subsamples, or the performance of human evaluators on the shift, which is typically plotted as $y = x$ for natural distribution shifts; typically this follows a *linear* trend.

Finally, we leverage **Effective Robustness Ratio (E.R.R.)** (Feuer et al., 2022), computed as average shift accuracy over base task accuracy. We find that this is an effective measure when we limit our comparisons to models with similar base accuracy, but do not recommend its use for comparing models whose base accuracy differs substantially.

**VL-loss models are not always more robust than CE-loss models.** One of the major themes in recent research into distributional robustness has been the seemingly unmatched performance of VL

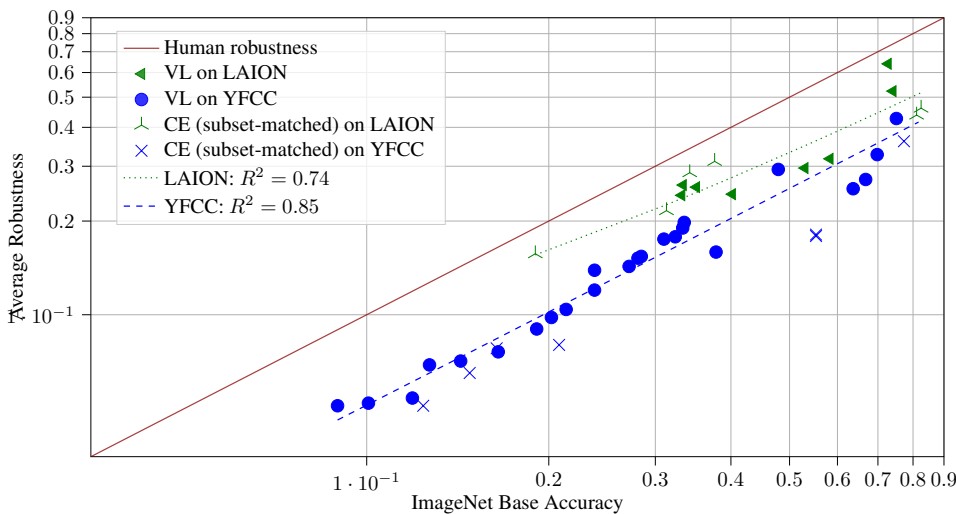

Figure 2: **Choice of loss function impacts effective robustness.** *In this figure, we display combined results from models trained on ImageNet-100 and ImageNet-1000. The $x$ axis shows accuracy on the base task, and the $y$ axis average performance under shift. We find that a CE-loss ResNet-50 with subset-matched labels equals or exceeds VL-loss robustness when pretraining on LAION data, but on YFCC, the inverse is true.*

models under distribution shift.Radford et al. (2021) Indeed, it is this performance which has driven much of the interest in these models.

However, in Fig. 2, we present a novel finding; depending on the underlying choice of dataset, CE-loss models using subset matching can actually be *more* robust than VL models. Specifically;

1. On LAION-15m and CC-12m, a simple subset matching labeling strategy produces CE-loss models which are sometimes more robust than VL models.
2. On YFCC, the same strategy produces CE-loss models which are always less robust than VL models.

These results hold at both low and high accuracies and are unaffected by subsampling the dataset. Additional experiments in Table 2 show that even when we hold the loss function and dataset size constant, changing label matching (filtration) strategy alone can affect distributional robustness by as much as 20 percent.

Table 2: **Subset matching strategy can impact effective robustness.** *Here, we compare models with our labels to models using the default ImageNet class labels. We find that under a strict subset matching strategy, our labels perform better than default labels and a random subsample of YFCC.*

| Dataset Size | Data Source | Technique | Labels Used | Strategy | Val. Acc. (ImageNet) | Avg. Rob. | E.R.R. |
|---|---|---|---|---|---|---|---|
| 2m | yfcc | VL | none | Subsample | $0.119 \pm 0.003$ | $0.054 \pm 0.002$ | 0.454 |
| 2.3m | yfcc | VL | default | Strict | $0.148 \pm 0.003$ | $0.087 \pm 0.002$ | 0.588 |
| 2.3m | yfcc | Subset matching | default | Strict | $0.124 \pm 0.003$ | $0.065 \pm 0.002$ | 0.526 |
| 2.2m | yfcc | VL | ours | Strict | $0.167 \pm 0.003$ | $0.108 \pm 0.002$ | 0.647 |
| 2.2m | yfcc | Subset matching | ours | Strict | $0.151 \pm 0.003$ | $0.084 \pm 0.002$ | 0.556 |

Our findings highlight a fundamental challenge in evaluating results on distributional robustness; researchers cannot rely on observations made on logit-transformed linear fits (as in Recht et al. (2019)) to hold both in the low- and high-accuracy regimes, unless model architecture, loss function, dataset, labeling strategy and evaluation metrics are fixed. Ratio-based measures such as the one described in Feuer et al. (2022) also cannot be used to effectively compare models whose base accuracy differs widely.

Table 3: *An overview of the four main datasets in CaptionNet.* **Label sources** *lists the source(s) for integer labels in the dataset.* **Caption sources** *lists the sources for captions in the dataset.* **Supervised** *indicates when ground truth labels exist for the dataset. CE-loss models benefit most from supervised data.* **Filtered** *indicates when the dataset contents were processed in some way prior to inclusion in the dataset. VL-loss models struggle on unfiltered data.* **Balanced** *indicates whether the dataset is approximately class-balanced. CE-loss models struggle on unbalanced data.*

| Dataset Name and Abbreviation | Label Sources | Caption Sources | Supervised | Filtered | Balanced |
|---|---|---|---|---|---|
| ImageNet-100 (in100) | Ground truth, subset matched | flickr, BLIP | Yes | Yes | Yes |
| OpenImages-100 (oi100) | Ground truth | flickr, BLIP, annotated | Yes | No | No |
| LAION-100 (laion100) | Subset matched | alt-text | No | Yes | No |
| YFCC-100 (yfcc100) | Subset matched | flickr | No | No | No |

The ideal comparison, then, would take place in a setting where both VL and CE-loss models were capable of achieving high base accuracy. Unfortunately, ImageNet does not provide such a setting; even VL models trained on the largest publicly available unsupervised dataset, LAION, have yet to match the performance of CLIP, and training even one such model is enormously demanding.

## 3 CAPTIONNET

To resolve this challenge, we introduce CaptionNet, a collection of four new training datasets designed to be evaluated on a subset of ImageNet. Each dataset in CaptionNet is either a subset or a superset of an existing dataset, and each is fully captioned and fully labeled, either using ground-truth or synthetic labels.

1. **ImageNet-100 (in100):** A superset of 100 ImageNet-Captions (Fang et al. (2022)) classes with over 50,000 new human-authored ground-truth labels, flickr-captions and blip-captions.
2. **OpenImages-100 (oi100):** A subset of the OpenImages (Kuznetsova et al. (2020)) dataset with restored original flickr-captions, and new BLIP-captions; samples selected by mapping human-labeled OpenImages-100 classnames to ImageNet-100 classnames.
3. **LAION-100 (laion100):** A subset of the unlabeled LAION (Schuhmann et al. (2021a)) dataset with samples selected via subset matching on ImageNet-100 classes.
4. **YFCC-100 (yfcc100):** A subset of the unlabeled YFCC dataset (Thomee et al. (2016)) with samples selected via subset matching on ImageNet-100 classes.

We compare some of the key properties of each component of CaptionNet in Table 3.

More information on the process used to create CaptionNet is available in Section F. Additional details on the size and composition of each CaptionNet subset can be found in Section 4.2.

**Training on CaptionNet.** In order to minimize differences in model architecture, we train two families of models: A ResNet-50 for CE-loss models, and a VL-loss model with a ResNet-50 vision backbone. The only difference in the two architectures is that for CE models, we append a ResNet-50 with a 1000-class linear head; we allow this since, as noted in (Radford et al., 2021; Santurkar et al., 2022), this does not seem to affect CLIP performance. In order to control for dataset size, we train models on various subsets of CaptionNet and measure base accuracy and distributional robustness.

CE-loss models are typically trained with full (32 bit floating point) precision with batch size of 128 and gradient clipping. VL models are typically trained with mixed precision (32-bit for gradients, and 16-bit for weights), for a batch size of 256, and do not use gradient clipping. Models are typically distributed across a single node with 4 NVIDIA GPUs; our largest models were trained on 16 NVIDIA GPUs. We use the AMP library to implement the training process. We train our larger models for 32 or 64 epochs unless otherwise specified. For CaptionNet models, we train all models for 256 epochs. In the CaptionNet experiments, we also experimented with models that trained on a subset-matched dataset matching on all 1000 classes in ImageNet. We refer to these datasets as YFCC-2.2Mn (yfcc22m) and YFCC-3.9Mn (yfcc39m). These datasets were trained for 64 epochs owing to computational constraints.

**Evaluation on CaptionNet.** As is standard for this type of study, we validate on ImageNet-100 validation images (in100-val), irregardless of the choice of pretraining dataset. Nguyen et al. (2022)

We report the best ID/OOD accuracy score for each model, with best being determined by the model's peak performance on ImageNet-100 validation. For ImageNet-R and ImageNet-A, which are subsets of ImageNet, we evaluate only the 35 shared classes. Following Santurkar et al. (2022), we use SimCLR augmentations (resize, crop, flip, jitter, blur, grayscale) rather than CLIP augmentations (resize and crop) for all model training on CaptionNet. The codebase necessary to reproduce our experiments can be found in our repository.

## 4   RESULTS ON CAPTIONNET

Our main experiments, in Table 4 evaluated each subset of CaptionNet separately, followed by combinations of those subsets. Since prior experiments controlled for base accuracy, model architecture, dataset composition and dataset size, we focused on comparing VL-loss and CE-loss models. Separately, we also performed ablations on the effects of subset matching as a labeling strategy; those results can be found in Section E.

**ImageNet-100 performance.** In order to ensure that the baseline performance of VL and CE models is essentially comparable on ImageNet-100 and the standard ImageNet despite the newly added images, we train a VL (using "A photo of a $CLASSNAME" captions) and CE model from scratch on ImageNet-100 and compare; we find that evaluation accuracy of the smaller models is very similar to their larger counterparts; ImageNet-100 is a good proxy for ImageNet.

**OpenImages-100 performance.** The baseline performance of OpenImages-100 with CE-loss is considerably worse than ImageNet-100 with CE-loss, most likely because of the natural class imbalance in the dataset (arising from the fact that images were not filtered). The baseline performance of OpenImages-100 with VL-loss was highly dependent on captioning strategy; we discuss these results in Section 4.2.

**LAION-100, YFCC-100 performance.** CE-loss models failed to learn anything on the unsupervised datasets. VL-loss models learned nearly as much as they did from in100, and more than oi100, indicating that dataset size, rather than dataset composition or label accuracy, is the most important factor for VL-loss.

**ImageNet-100 + Supervised data.** ImageNet-100 + OpenImages-100 is about twice the size of ImageNet-100 alone, and every sample has a ground truth label. We find that in VL, this combination adds a small amount of distributional robustness compared to ImageNet-100 alone; in CE, we gain base accuracy and distributional robustness.

From this, we conclude that CE-loss models benefit more from adding supervised data than VL-loss. CE-loss models appear to be capable of learning very accurate representations with relatively little data; the base accuracy of the best-performing subset-matched model in Table 4, trained on fewer than 1 million samples, was higher than the best VL model, which trained on over 400 million samples.

**ImageNet-100 + Unsupervised data.** When we scale up on noisily supervised data, we increase the dataset between 4x and 10x its original size. In this paradigm, we find that VL-loss models improve fairly smoothly with data scale, whereas CE-loss models improve most on the cleaner laion-100 data.

**ImageNet-100 + Both.** What happens when we combine all of our supervised data and the cleanest source of unsupervised data? VL performance actually degrades slightly; this result is quite surprising, indicating again that scale matters more than anything else in VL. CE-loss, on the other hand, improves in distributional robustness without losing base accuracy, and comes close to rivaling CLIP, trained on 400m samples.

**Training on out of distribution classes.** To test the effects of scaling up on out of distribution classes, our final set of experiments on CaptionNet involved training on ImageNet-100 + YFCC-3.9Mn and ImageNet-100 + YFCC-2.2Mn. These datasets were filtered by subset-matching on 1000 ImageNet-classes; therefore, the vast majority of the labels in these datasets map to classes which are not evaluated by ImageNet-100 validation.

We train on the original captions for VL models, and subset matched labels for CE models and find vast discrepancies in performance. The VL model improves in both distributional robustness and accuracy, and the CE-loss model gets worse. This fits with a larger finding; *VL-loss models perform better when new tokens are seen, even if those tokens are not used during evaluation.* To verify this,

in Table 6, we train a model on ImageNet-100 in which we convert all tokens which are not in the prompt or classnames to 0. In this situation, we find that both accuracy and distributional robustness decline substantially. We see the same effect when we train VL-loss models on ImageNet-100 + YFCC3.9Mn, and improves disproportionately under shift.

Table 4: **A direct comparison of VL and CE-loss models on CaptionNet datasets highlights the importance of the loss function.** *The best-performing VL model is marked in* **boldface**, *and the best performing CE model is in* italics. *CE-loss models are supervised using ground-truth labels where they are available, and subset-matched labels where ground-truth labels are unavailable. VL-loss models are supervised using the best available captions for the dataset. We find that unfiltered datasets with only subset-matched labels are unlearnable on their own, but when blended them with ground-truth supervised data, the resulting model is more accurate and more robust than a VL model training on the same data with captions. VL distributional robustness is proportionately higher in low and medium regimes, but base task accuracy improves only after large amounts of additional data are added.*

| Dataset | Size | VL Val. Acc. (in100) | VL Avg. Rob. | VL E.R.R. | CE Val. Acc. (in100) | CE Avg. Rob. | CE E.R.R. |
|---|---|---|---|---|---|---|---|
| in100 | 124k | $0.574 \pm 0.014$ | $0.217 \pm 0.008$ | 0.378 | $0.801 \pm 0.011$ | $0.333 \pm 0.008$ | 0.416 |
| oi100 | 135k | $0.283 \pm 0.012$ | $0.131 \pm 0.008$ | 0.464 | $0.571 \pm 0.011$ | $0.274 \pm 0.008$ | 0.48 |
| yfcc100 | 375k | $0.378 \pm 0.013$ | $0.159 \pm 0.008$ | 0.421 | 0 | 0 | 0 |
| laion100 | 454k | $0.402 \pm 0.014$ | $0.244 \pm 0.008$ | 0.607 | 0 | 0 | 0 |
| in100+oi100 | 259k | $0.575 \pm 0.014$ | $0.251 \pm 0.008$ | 0.437 | $0.834 \pm 0.01$ | $0.395 \pm 0.009$ | 0.474 |
| in100+laion100 | 578k | $0.583 \pm 0.014$ | $0.317 \pm 0.008$ | 0.544 | $0.811 \pm 0.01$ | $0.438 \pm 0.009$ | 0.54 |
| in100+yfcc100 | 499k | $0.637 \pm 0.014$ | $0.254 \pm 0.008$ | 0.399 | $0.773 \pm 0.011$ | $0.361 \pm 0.008$ | 0.467 |
| *in100+oi100+laion100* | *713k* | $0.53 \pm 0.014$ | $0.296 \pm 0.008$ | 0.558 | $0.833 \pm 0.01$ | $0.546 \pm 0.009$ | *0.655* |
| in100+yfcc100+laion100 | 953k | $0.62 \pm 0.014$ | $0.322 \pm 0.008$ | 0.519 | $0.817 \pm 0.01$ | $0.467 \pm 0.009$ | 0.572 |
| in100+yfcc22m | 2.4M | $0.668 \pm 0.013$ | $0.272 \pm 0.008$ | 0.407 | $0.553 \pm 0.013$ | $0.181 \pm 0.008$ | 0.327 |
| in100+yfcc39m | 4.1M | $0.698 \pm 0.013$ | $0.327 \pm 0.008$ | 0.468 | $0.553 \pm 0.013$ | $0.179 \pm 0.008$ | 0.324 |
| yfcc15m | 14.8M | $0.751 \pm 0.013$ | $0.427 \pm 0.009$ | 0.569 | N/A | N/A | N/A |
| laion15m | 13.7M | $0.741 \pm 0.012$ | $0.523 \pm 0.009$ | 0.706 | N/A | N/A | N/A |
| **CLIP-WiT400m** | **400M** | $\mathbf{0.9} \pm 0.011$ | $\mathbf{0.707} \pm 0.008$ | 0.786 | N/A | N/A | N/A |
| ImageNet | 1.2M | N/A | N/A | N/A | $0.736 \pm 0.011$ | $0.238 \pm 0.008$ | 0.323 |

## 4.1 LABEL NOISE

Caption supervision can be very noisy; how do VL and CE-loss compare in their handling of noise? In Table E.2, we compare the accuracy of subset matched labels to ground truth labels on OpenImages-100, we find that even the best subset matching techniques apply correct labels to less than 10 percent of the available data, suggesting that the caption text and the ground truth labels have very little overlap in this dataset. In this high-noise setting, the VL-loss model is able to reach 20 percent zero-shot accuracy with flickr-captions, whereas a subset-matched CE-loss model (not listed in the table) does not learn at all. When we add synthetic captions to OpenImages-100, the VL-loss model improves to nearly 30 percent.

Just how sensitive are CE-loss models to dataset noise? In Table 5, we see that performance degrades significantly when we train using subset matching techniques and 10 percent of the labels are noisy, even when we control for the difference in dataset size. We also observe that not all noise is created equal; two models with similar error rates can have very different performance, depending on what those errors are. We discuss this surprising result in greater detail in Section E.4.

Table 5: **Not all errors are created equal; CLIP labels outperform subset matched labels on ImageNet-100.** *This table compares six CE-loss models with different labeling strategies. Abbreviations are defined in Section 1. We find that labels generated by a ViT-L CLIP model perform better on ImageNet-100 (in100) than subset matched labels, even though the true accuracy (which we determine by comparing predicted to ground truth labels) of each labeling method is very similar. Label accuracy and match count cannot fully explain differences in model performance on a dataset.*

| Label Source | True Acc. | Ds. Util. | Val. Acc. (in100) | in100-a | in100-r | in100-s | in100-v2 | Avg. Rob. | E.R.R. |
|---|---|---|---|---|---|---|---|---|---|
| Ground-truth | 1 | 1 | $0.801 \pm 0.012$ | 0.094 | 0.23 | 0.297 | 0.71 | $0.333 \pm 0.008$ | 0.416 |
| ViT-L CLIP | 0.9 | 1 | $0.778 \pm 0.012$ | 0.099 | 0.225 | 0.286 | 0.66 | $0.318 \pm 0.008$ | 0.409 |
| Subset Matched Size Ctrl. (oai-ttd-sc) | 0.89 | 1 | $0.714 \pm 0.013$ | 0.074 | 0.184 | 0.223 | 0.606 | $0.272 \pm 0.008$ | 0.381 |
| Subset Matched (oai-ttd-sc) | 0.89 | 0.72 | $0.651 \pm 0.013$ | 0.065 | 0.179 | 0.194 | 0.537 | $0.244 \pm 0.008$ | 0.375 |
| Subset Matched (oai-tags-sc) | 0.87 | 0.58 | $0.608 \pm 0.014$ | 0.065 | 0.147 | 0.168 | 0.506 | $0.222 \pm 0.008$ | 0.365 |
| Subset Matched (oai-title-sc) | 0.94 | 0.49 | $0.579 \pm 0.014$ | 0.062 | 0.154 | 0.168 | 0.47 | $0.214 \pm 0.008$ | 0.37 |

Table 6: **Eliminating all tokens not used in evaluation reduces evaluation accuracy.** *VL models perform worse on validation and under shift when all tokens which are not in the prompt or classnames are mapped to 0, showing that VL models learn about evaluation classes from non-evaluation classes.*

| Model | Val. Acc. (in100) | Avg. Rob. | E.R.R. |
|---|---|---|---|
| ImageNet-100 Flickr | $0.574 \pm 0.014$ | $0.217 \pm 0.008$ | 0.378 |
| *ImageNet-100 Token Stripped* | *0.476* $\pm 0.014$ | *0.177* $\pm 0.008$ | *0.372* |

## 4.2 CAPTIONING STRATEGY CAN AFFECT DISTRIBUTIONAL ROBUSTNESS

Although the focus of our experiments was loss functions and labeling strategies, we also wanted to briefly address the difference of opinion in recent works on the importance of caption contents, which we describe in detail in Section 1.

In our experiments on CaptionNet, which can be found in Table 11, we found that on ImageNet-100, there was no change in performance whatsoever when using BLIP+Title captioning. On OpenImages-100, model performance improved by 10% when using human-annotated+BLIP+Title captioning. The impact of synthetic captions, then, seems to be strongest when the pretraining dataset is labeled but unfiltered, suggesting that the machine captions act as a kind of pseudo machine-label.

**What makes a caption more robust?** In Table G.1, we observe that by many common measures, robust and non-robust captions are very similar; to the extent they are dissimilar, the measures would seem to favor the non-robust YFCC captions rather than the robust LAION captions. Measures such as length, language of origin (as estimated by a common Python language detection package), or token diversity are unlikely to explain differences in model performance.

## 5 DISCUSSION AND RECOMMENDATIONS

Caption supervision enables distributionally robust model training. It primarily does so by indirect means; captions serve as a dataset filtration method which are interpreted as labels by the model. However, in the VL-loss paradigm, the contents of the captions themselves can also affect both accuracy and distributional robustness, and not necessarily in equal proportion.

1. **Loss function matters.** In Section 2, we showed that the choice of VL or CE-loss can have a substantial effect on model distributional robustness and base accuracy, even when dataset sizes are similar.
2. **CE learns best when noise is low and classes are few.** In Section 4 and its subsections, we showed that CE-loss models learn high-accuracy, robust representations from compact datasets, but are highly sensitive both to the amount and to the type of noise present in the labels. CE-loss models also tend to perform worse on in-distribution classes when training on out-of-distribution classes (classes which are not evaluated).
3. **VL learns best when lots of data is available for many classes.** Also in Section 4, we showed that VL-loss models tend to be more robust at low accuracy, and can learn a basic representation of a very large number of classes even when labels are less than 10 percent accurate; however, they do not benefit as much from ground-truth captions as CE-loss models benefit from ground-truth labels. Furthermore, they require far more data to attain high accuracy.
4. **Image quality trumps caption quality.** In Section 4.2, we showed that descriptive caption supervision can result in better performance when caption quality is low, but has no effect on accuracy or distributional robustness when caption quality is high.

**Future research directions.** Our controlled study has helped illuminate some of the meaningful differences between CE-loss and VL-loss models. We believe that research into how to shore up the weaknesses of either approach, or discovery of a new method which blends their strengths, would be a useful direction for future efforts.

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

# A    DISTRIBUTION SHIFTS ON IMAGENET

ImageNet is a large-scale visual ontology of images built upon the backbone of the WordNet structure. ImageNet aims to populate the majority of the 80,000 synsets of WordNet with an average of 500–1000 clean and full resolution images, making it a roughly class-balanced, fully supervised dataset. Deng et al. (2009)

There now exist a wide range of distribution shifts on ImageNet. These are novel test datasets designed to overcome some of the limitations of the original benchmark. While they cannot remedy issues with the labeling scheme, these datasets do provide challenging new contexts in which to analyze classifier performance.

ImageNet-V2 was designed to duplicate, as closely as possible, the original ImageNet test set. It was intended to answer the question of whether ImageNet-trained classifiers could successfully generalize even to the most mild of distribution shifts.Recht et al. (2019)

Imagenet-Sketch is a distribution shift covering sketches, paintings, drawings and illustrations of ImageNet classes. This test set is very large and comprehensive.Wang et al. (2019)

Imagenet-R is a 200-class subset of ImageNet-2012 focused on renditions of everyday objects, defined broadly as drawings, paintings, photographs of food art, etc.Hendrycks et al. (2021a)

Imagenet-A is a 200-class subset of ImageNet-2012 which was algorithmically selected – the natural distribution shift captured here is the set of ImageNet-class images which most often fool a RN50. This test is challenging, and tends to include a lot of images with challenges such as occlusion, changes in angle or position, and changes in brightness.Hendrycks et al. (2021b)

## A.1    DIFFERENT SHIFTS RESPOND TO DIFFERENT INTERVENTIONS

Recent works such as Fang et al. (2022) demonstrate the power of effective robustness as an explanatory tool for performance differences in VL models; Miller et al. (2021) showed that there exists a strong correlation between most models trained on random subsets of a data distribution, and the fully trained model. However, these authors also caution that it has significant limitations – Taori et al. (2020) and Nguyen et al. (2022) show that models trained on more (or different data) can significantly change the effective robustness line of a particular model, and also that these changes were shift-specific, with stronger fits on shifts like ImageNet-V2 and weaker fits on shifts like ImageNet-A. In Section C, we extend these findings to almost 1000 publicly available vision models, including almost 100 VL models, demonstrating, in granular detail, nonlinearities that appear in many common shifts as models train on larger datasets and at higher accuracies.

While it is tempting to deal with distribution shifts as a kind of monolith, the truth is, perhaps unsurprisingly, more complex.

We found that ImageNet-V2 seemed to respond more to model architecture than other shifts, with the handful of non-ResNet models we evaluated outperforming nearly all other models, regardless of training objective.

ImageNet-R and ImageNet-Sketch both showed high sensitivity to the training data, with the CC12M and LAION-15m distributions considerably outperforming even the best YFCC-trained models. These types of shifts are particularly amenable to subset matching strategies.Fig. 6, Fig. 4

On ImageNet-A, CE models significantly underperformed compared to VL models regardless of the data, and all models significantly underperformed compared to the ViT-L CLIP.Fig. 5

We also note that there is no readily apparent logit-scaled linear trend in these distribution shifts when one considers models trained on a wide range of different datasets, underscoring the importance of a well-chosen baseline for comparison.

We find that different shifts tend to disadvantage different kinds of models, which makes improving on all of them simultaneously very challenging. The fact that ViT-L CLIP was able to do is both impressive and, given the vital importance of the underlying data distribution in such measures, a mystery which is unlikely to ever be solved. Even the massive public datasets such as LAION are unable to match the performance of the dataset CLIP was trained on, although other factors might possibly have played a role.

A standardized benchmark of distribution shifts on ImageNet would be a welcome contribution to this area of research.

## B  PRETRAINING DATASETS

Today, many SOTA models are pretrained on web-scale unsupervised data. We utilized three such datasets in our experiments. We observe that one major challenge of conducting research on unsupervised datasets is that the links provided as part of the dataset fail more and more over time, leading to each group getting a different version of the dataset. Therefore, to the extent possible, we report the details of each dataset in the appendix, and encourage other researchers working with these datasets to do the same.

CC-12M is a lightly supervised web-scale dataset created by Google. The image-caption pairs in CC-12M were filtered and selected for the purposes of training models to caption images.Changpinyo et al. (2021) Our version of CC12M contained 9703885 image-caption pairs.

YFCC-15M is a subset of YFCC-100M, which is 100M image-metadata pairs taken from Yahoo-Flickr in 2016. The subset was selected by OpenAI. This dataset contains images and metadata, which includes a "title" and a "description" field. These fields are combined and processed in various ways by researchers in order to generate captions for models to train on.Thomee et al. (2016) Our version of YFCC contained 14825134 image-caption pairs.

LAION is a 5B image-caption dataset recently created by LAION.ai. It is the first publicly available dataset which matches the scale of the datasets used by the large companies to train their best models.Schuhmann et al. (2021a) The subset of LAION we refer to as LAION-15m contained 13775512 image-caption pairs.

## C  LARGE-SCALE EVALUATION OF CE VS VL DISTRIBUTIONAL ROBUSTNESS

In order to get a more complete picture of the current landscape of model distributional robustness, we evaluated nearly 1000 models on our suite of distribution shifts, including all of the models with metrics reported by Taori et al. (2020); Wightman (2019); Feuer et al. (2022), models trained with LiT and Wise-FT objectives as described in Wortsman et al. (2021); Zhai et al. (2021) and all of the models trained for this paper. The results are shown in Fig. C. We find that in the very high accuracy regime, the logit-transformed linear fit of CE models fails to hold, and CE distributional robustness increases faster than predicted, approaching VL distributional robustness.

## D  COMPLETE TABLE OF MODEL RESULTS

In Table D and Table D, we present the collected results of all of our experiments, both on CaptionNet and on larger portions of YFCC, LAION and CC12M.

## E  SUBSET MATCHING STRATEGIES

In this section, we define and describe some important variations on the basic subset matching strategy as described in the main paper. All of our subset matching experiments utilized one of three matching strategies;

**Strict:** Only match on samples which have exactly one class in them. Strict matching degrades as matching strategies grow more aggressive, and is also strongly affected by the number of classes evaluated; however, it tends to be the least noisy method, making it useful in some contexts.

**Single class:** greedily take the first matching class as the true class and ignore all others. As a general matter, we found that single-class matching struck the best balance between dataset utilization and accuracy.

**Multi class:** Match on all matching classes, up to 25 classes per sample. We found that this approach tended to decrease accuracy, and that under most strategies, multiclass matches were uncommon anyway.

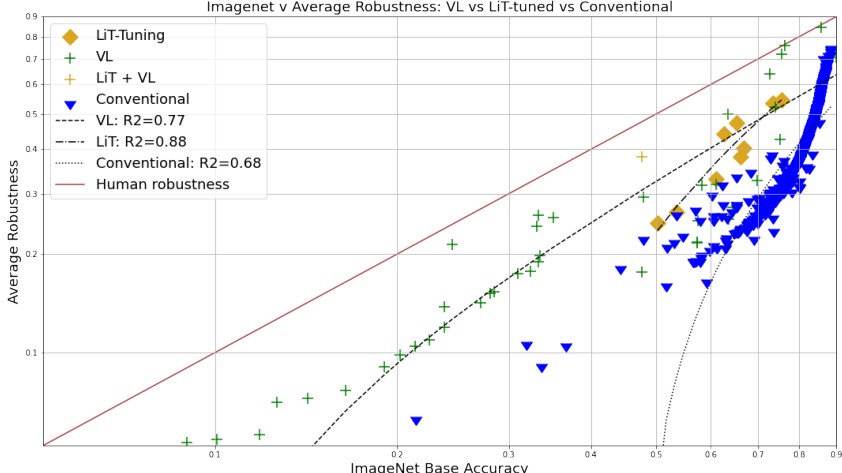

Figure 3: **The new picture of effective robustness.** *This plot, which contains inference results from nearly 1000 models, shows a more complicated landscape of effective robustness than previous investigations. Ablations on the caption space show that even when the information in captions is aggressively transformed or reduced, effective robustness is preserved. In the low-accuracy regime, RN50s trained on integer-captioned yfcc and LAION data are able to match or exceed VL effective robustness. In the high accuracy regime, LiT-tuned VL models and wise-ft models approach, but do not reach, the VL-robust line. CE loss models also approach the VL line at very high base accuracies.*

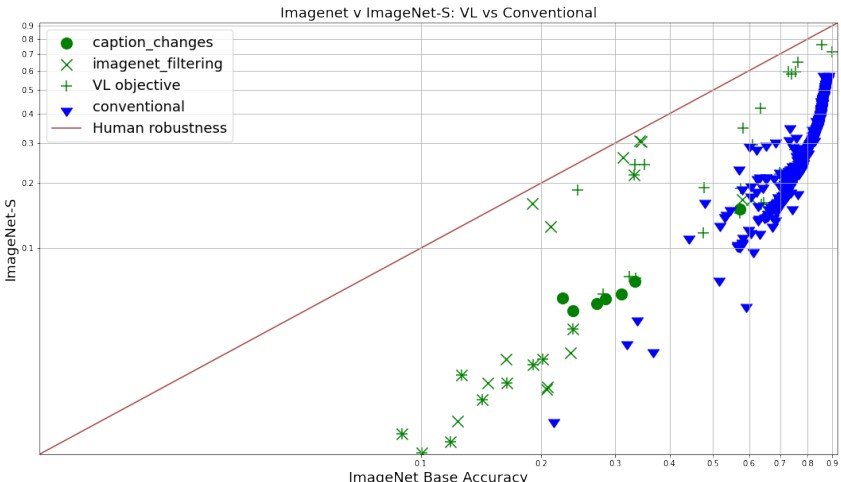

Figure 4: **Non-linearities in ImageNet-Sketch.** *ImageNet-sketch performance is not linear, with only the very largest VL models showing a reliable improvement over CEly trained models, when controlling for dataset size.*

In addition to differences in matching strategy, we also experimented with different sets of terms for matching, which we refer to as **OpenAI** or **in1k_openai** (from Radford et al. (2021)), **default** or **in1k_default** (from Fox, E., and Guestrin, C. (n.d.). Coursera Machine Learning Specialization), and **ours** or **in1k_ours**, created by us.

All of these labels were chosen heuristically, to the best of our knowledge. We found that the heuristic changes to term matches often had substantial effect on accuracy; however, we leave the algorithmic discovery of optimal subset matching terms to future work.

We have released all of our matching terms in VLHub (EG: /vlhub/metadata/in1k_ours.txt).

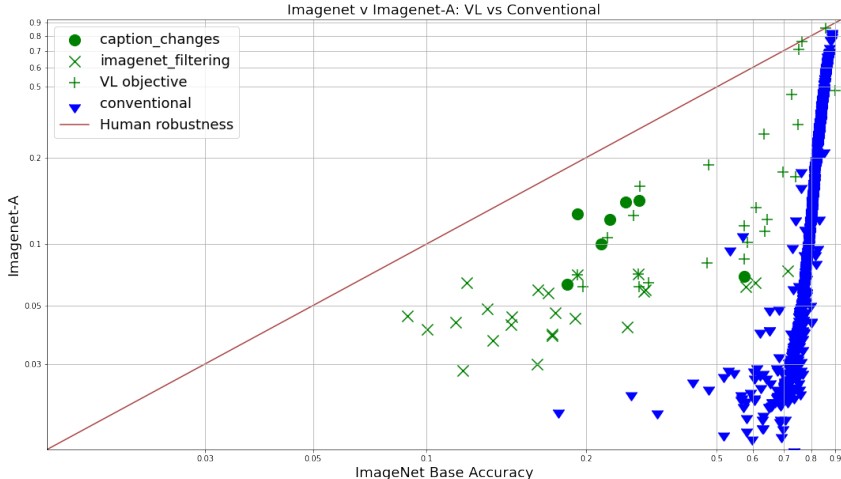

Figure 5: **ImageNet-A is learnable by all models at extremely high base accuracy.** *Although VL models seem to learn ImageNet-A faster than CE models, CE models reach near-parity with VL models when base accuracy gets very high.*

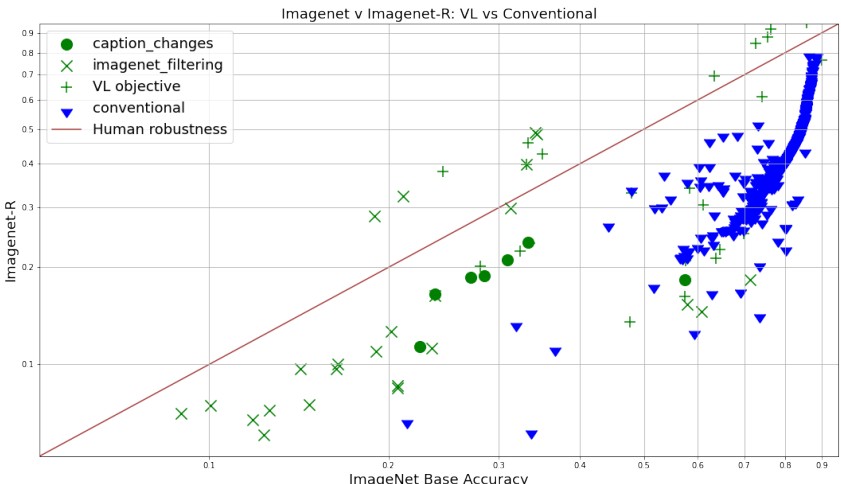

Figure 6: **VL performance on ImageNet-R outstrips base accuracy.** *On ImageNet-R, which is a 200-class subset of ImageNet, VL models are able to achieve higher accuracy than on ImageNet itself. VL continues to outperform CE models on this dataset, even at very high accuracies.*

### E.1 DATASET UTILIZATION

Dataset utilization is defined in Section 1. We define accuracy as the number of correctly classified samples over correctly plus incorrectly classified samples.

Because ImageNet-100 and OpenImages-100 have ground truth labels, the optimal model would have a dataset utilization of 1.0 (100 percent).

### E.2 PERFORMANCE OF SUBSET MATCHING STRATEGIES ON IMAGENET-100 AND OPENIMAGES-100

When ground-truth labels do not exist, it is impossible to be certain how accurate any labeling strategy is. Therefore, we take advantage of the existence of ground-truth labels for OpenImages-100 and ImageNet-100 to compute the accuracy of a wide range of strategies.

Table 7: **Complete results table: non-CaptionNet models.** *This table includes results for all of the larger models in our study, including performance broken down by specific distribution shift.*

| approx_samples | dataset | technique | matching terms | strategy | in-val | in-a | in-r | in-s | in-v2 | Avg. Rob. | E.R.R. |
|---|---|---|---|---|---|---|---|---|---|---|---|
| 14.8m | yfcc-15m | VL | None | Baseline | 0.324 | 0.136 | 0.223 | 0.073 | 0.280 | 0.178 | 0.55 |
| 4m | yfcc-15m | VL | None | Subsample | 0.191 | 0.060 | 0.110 | 0.026 | 0.165 | 0.09 | 0.471 |
| 2m | yfcc-15m | VL | None | Subsample | 0.119 | 0.041 | 0.066 | 0.010 | 0.100 | 0.054 | 0.454 |
| 2.3m | yfcc-15m | VL | default | Strict | 0.143 | 0.048 | 0.097 | 0.017 | 0.123 | 0.071 | 0.497 |
| 2.3m | yfcc-15m | subset matched | default | Strict | 0.124 | 0.023 | 0.059 | 0.013 | 0.107 | 0.051 | 0.411 |
| 3.9m | yfcc-15m | VL | default | Multiclass | 0.238 | 0.071 | 0.164 | 0.040 | 0.206 | 0.12 | 0.504 |
| 3.9m | yfcc-15m | subset matched | default | Multiclass | 0.208 | 0.034 | 0.085 | 0.021 | 0.180 | 0.08 | 0.385 |
| 2.2m | yfcc | VL | ours | Strict | 0.165 | 0.044 | 0.100 | 0.021 | 0.138 | 0.076 | 0.461 |
| 2.2m | yfcc | subset matched | ours | Strict | 0.148 | 0.033 | 0.074 | 0.021 | 0.130 | 0.065 | 0.439 |
| 3.1m | yfcc | VL | ours | Multiclass | 0.202 | 0.058 | 0.127 | 0.028 | 0.180 | 0.098 | 0.485 |
| 3.1m | yfcc | subset matched | ours | Multiclass | 0.164 | 0.040 | 0.097 | 0.028 | 0.145 | 0.078 | 0.476 |
| 10m | yfcc | VL | NOT ours & default | Anticlass | 0.127 | 0.065 | 0.071 | 0.023 | 0.118 | 0.069 | 0.543 |
| 13.7m | LAION | VL | None | Baseline | 0.351 | 0.065 | 0.425 | 0.241 | 0.294 | 0.257 | 0.733 |
| 8m | LAION | subset matched | openai | Strict | 0.342 | 0.059 | 0.489 | 0.308 | 0.289 | 0.286 | 0.836 |
| 32m | LAION | subset matched | openai | Strict | 0.376 | 0.067 | 0.524 | 0.333 | 0.319 | 0.311 | 0.827 |
| 2.6m | LAION | VL | openai | Singleclass | 0.332 | 0.072 | 0.398 | 0.217 | 0.281 | 0.242 | 0.73 |
| 2.6m | LAION | subset matched | openai | Singleclass | 0.313 | 0.039 | 0.299 | 0.259 | 0.268 | 0.216 | 0.69 |
| 9.7m | cc12m | VL | openai | Baseline | 0.245 | 0.062 | 0.380 | 0.186 | 0.228 | 0.214 | 0.873 |
| 2.4m | cc12m | VL | openai | Singleclass | 0.189 | 0.047 | 0.293 | 0.126 | 0.165 | 0.158 | 0.836 |
| 2.4m | cc12m | subset matched | openai | Singleclass | 0.211 | 0.046 | 0.324 | 0.127 | 0.182 | 0.17 | 0.806 |

Table 8: **Complete Results table: CaptionNet models.** *This table includes results for all of the CaptionNet models in our study, including performance broken down by specific distribution shift. VL models perform better on OpenImages when flickr-captions are replaced with synthetic captions (BLIP+Title captions), but the same captioning method provides no benefits on ImageNet-100.*

| name | dataset | technique | in-val | in-a | in-r | in-s | in-v2 | Avg. Rob. | E.R.R. |
|---|---|---|---|---|---|---|---|---|---|
| in100-vl | in100 | VL | 0.574 | 0.085 | 0.163 | 0.146 | 0.474 | 0.217 | 0.378 |
| in100-sup | in100 | CE | 0.801 | 0.094 | 0.230 | 0.297 | 0.710 | 0.333 | 0.416 |
| oi100-vl-bestcaps | oi100 | VL | 0.225 | 0.064 | 0.114 | 0.057 | 0.203 | 0.11 | 0.489 |
| oi100-sup | oi100 | CE | 0.571 | 0.113 | 0.233 | 0.237 | 0.512 | 0.274 | 0.48 |
| in100+oi100-vl | in100+oi100 | VL | 0.575 | 0.122 | 0.209 | 0.189 | 0.484 | 0.251 | 0.437 |
| in100+oi100-sup | in100+oi100 | CE | 0.834 | 0.135 | 0.324 | 0.384 | 0.738 | 0.395 | 0.474 |
| in100+laion100-vl | in100+laion100 | VL | 0.583 | 0.102 | 0.342 | 0.348 | 0.476 | 0.317 | 0.544 |
| in100+laion100-subset matched | in100+laion100 | subset matched | 0.811 | 0.129 | 0.418 | 0.478 | 0.725 | 0.438 | 0.54 |
| in100+yfcc100+laion100-vl | in100+laion100+yfcc100 | VL | 0.620 | 0.152 | 0.301 | 0.291 | 0.544 | 0.322 | 0.519 |
| in100+yfcc100+laion100-subset matched | in100+laion100+yfcc100 | subset matched | 0.817 | 0.202 | 0.449 | 0.475 | 0.740 | 0.467 | 0.572 |
| in100+yfcc22m-vl | in100+yfcc15m | VL | 0.668 | 0.142 | 0.205 | 0.172 | 0.567 | 0.272 | 0.407 |
| in100+yfcc22m-subset matched | in100+yfcc15m | subset matched | 0.553 | 0.047 | 0.114 | 0.119 | 0.443 | 0.181 | 0.327 |
| in100+yfcc39m-vl | in100+yfcc15m | VL | 0.698 | 0.216 | 0.252 | 0.220 | 0.620 | 0.327 | 0.468 |
| in100+yfcc39m-subset matched | in100+yfcc15m | subset matched | 0.553 | 0.062 | 0.111 | 0.101 | 0.442 | 0.179 | 0.324 |
| in1k-1.2m-in100 | in1k | CE | 0.736 | 0.017 | 0.144 | 0.187 | 0.605 | 0.238 | 0.323 |
| in100+yfcc100-vl | in100+yfcc100 | VL | 0.637 | 0.115 | 0.213 | 0.160 | 0.526 | 0.254 | 0.399 |
| in100+yfcc100-subset matched | in100+yfcc100 | subset matched | 0.773 | 0.166 | 0.293 | 0.309 | 0.675 | 0.361 | 0.467 |
| in100-subset matched-title | in100 | subset matched | 0.579 | 0.062 | 0.154 | 0.168 | 0.470 | 0.214 | 0.37 |
| in100-subset matched-ttd-size 256ep | in100 | subset matched | 0.714 | 0.074 | 0.184 | 0.223 | 0.606 | 0.272 | 0.381 |
| in100-subset matched-tags | in100 | subset matched | 0.608 | 0.065 | 0.147 | 0.168 | 0.506 | 0.222 | 0.365 |
| in100-subset matched-ttd | in100 | subset matched | 0.651 | 0.065 | 0.179 | 0.194 | 0.537 | 0.244 | 0.375 |
| in100-vl-blip | in100 | VL | 0.574 | 0.070 | 0.184 | 0.153 | 0.458 | 0.216 | 0.376 |
| in100-tokenstrip | in100 | VL | 0.476 | 0.081 | 0.136 | 0.118 | 0.372 | 0.177 | 0.372 |
| yfcc15m-in100 | yfcc15m | VL | 0.751 | 0.349 | 0.371 | 0.282 | 0.704 | 0.427 | 0.569 |
| laion15m-in100 | laion | VL | 0.741 | 0.205 | 0.611 | 0.582 | 0.693 | 0.523 | 0.706 |
| CLIP-WIT400m-in100 | wit | VL | 0.900 | 0.482 | 0.764 | 0.713 | 0.867 | 0.707 | 0.786 |
| oi100-blipcaption | oi100 | VL | 0.283 | 0.066 | 0.137 | 0.095 | 0.227 | 0.131 | 0.464 |
| oi100-flickrcaption | oi100 | VL | 0.197 | 0.061 | 0.098 | 0.048 | 0.172 | 0.095 | 0.482 |
| in100+oi100+laion100-vl | in100+laion100+oi100 | VL | 0.530 | 0.092 | 0.297 | 0.327 | 0.468 | 0.296 | 0.558 |
| yfcc100-vl | yfcc100 | VL | 0.378 | 0.107 | 0.108 | 0.082 | 0.337 | 0.159 | 0.421 |
| laion100-vl | laion100 | VL | 0.402 | 0.092 | 0.274 | 0.245 | 0.363 | 0.244 | 0.607 |
| in100-cliplabels | in100 | CE | 0.778 | 0.099 | 0.225 | 0.286 | 0.660 | 0.318 | 0.409 |
| in100+oi100+laion100-subset matched | in100+laion100+oi100 | subset matched | 0.825 | 0.152 | 0.448 | 0.491 | 0.754 | 0.461 | 0.559 |

In Table E.2, we list exact accuracy results for a range of subset matching strategies by comparing them directly to the ground truth labels.

### E.3   PERFORMANCE OF VARIATIONS ON SIMPLE SUBSET MATCHING

There are many ways one could conceivably improve on the simple strategies described above. We explore some of them and describe them here.

Fuzzy subset matching uses the Levenshtein distance between tokens to locate potential matches. We used the standard Python library fuzzywuzzy with a setting of 55 (higher approaches found few matches). We found that this approach was considerably slower than simple subset matching, but offered only limited benefits.

Table 9: **Complete Results table: Subset Matching Strategies.** *This table contains direct comparisons of many of the subset matching strategies evaluated in our study; we consider a range of metrics, in particular raw accuracy and dataset utilization. TotalMatch is a count of how many samples were matched to some label by MC matching. MaxCorrS is count of how many samples were correctly matched to some label by the most successful strategy (typically SC). MaxDU is a measure of the highest dataset utilization possible using any strategy (typically SC).*

| Name | TotalMatch | MC Acc | SC Acc | Strict Acc | MaxCorrS | MaxDU |
|---|---|---|---|---|---|---|
| in100-tags-ours | 88597 | 0.76 | 0.86 | 0.92 | 76193 | 0.61 |
| in100-tags-openai | 83327 | 0.79 | 0.87 | 0.92 | 72494 | 0.58 |
| in100-tags-openai-synset | 99708 | 0.31 | 0.58 | 0.78 | 57831 | 0.47 |
| in100-tags-default | 93982 | 0.74 | 0.84 | 0.91 | 78945 | 0.63 |
| in100-descr-ours | 37685 | 0.7 | 0.8 | 0.83 | 30148 | 0.24 |
| in100-descr-openai | 35259 | 0.75 | 0.83 | 0.86 | 29265 | 0.24 |
| in100-descr-openai-synset | 51148 | 0.27 | 0.48 | 0.61 | 24551 | 0.2 |
| in100-descr-default | 40786 | 0.66 | 0.78 | 0.82 | 31813 | 0.26 |
| in100-title-ours | 69637 | 0.91 | 0.93 | 0.95 | 64762 | 0.52 |
| in100-title-openai | 64919 | 0.91 | 0.94 | 0.95 | 61024 | 0.49 |
| in100-title-openai-synset | 76299 | 0.39 | 0.76 | 0.87 | 57987 | 0.47 |
| in100-title-default | 74219 | 0.88 | 0.92 | 0.94 | 68281 | 0.55 |
| in100-titletags-ours | 103655 | 0.77 | 0.9 | 0.93 | 93290 | 0.75 |
| in100-titletags-openai | 97181 | 0.8 | 0.9 | 0.93 | 87463 | 0.7 |
| in100-titletags-default | 109192 | 0.75 | 0.88 | 0.92 | 96089 | 0.77 |
| in100-ttd-ours | 107559 | 0.73 | 0.89 | 0.92 | 95728 | 0.77 |
| in100-ttd-openai | 101210 | 0.77 | 0.89 | 0.92 | 90077 | 0.72 |
| in100-ttd-default | 113175 | 0.69 | 0.87 | 0.9 | 98462 | 0.79 |
| oi100-tags-ours | 19880 | 0.45 | 0.49 | 0.51 | 9741 | 0.07 |
| oi100-tags-openai | 17182 | 0.53 | 0.56 | 0.58 | 9622 | 0.07 |
| oi100-tags-default | 20371 | 0.51 | 0.56 | 0.58 | 11408 | 0.08 |
| oi100-descr-ours | 10286 | 0.32 | 0.37 | 0.35 | 3806 | 0.03 |
| oi100-descr-openai | 8600 | 0.4 | 0.44 | 0.42 | 3784 | 0.03 |
| oi100-descr-default | 12505 | 0.28 | 0.35 | 0.31 | 4377 | 0.03 |
| oi100-title-ours | 16437 | 0.54 | 0.56 | 0.51 | 9205 | 0.07 |
| oi100-title-openai | 14999 | 0.61 | 0.63 | 0.58 | 9449 | 0.07 |
| oi100-title-default | 18975 | 0.53 | 0.56 | 0.48 | 10626 | 0.08 |
| oi100-title-openai-fuzzy | 15492 | 0.56 | 0.61 | 0.59 | 9450 | 0.07 |
| oi100-title-openai-synset | 33685 | 0.24 | 0.29 | 0.3 | 9769 | 0.07 |
| oi100-titletags-default | 31440 | 0.47 | 0.53 | 0.52 | 10319 | 0.08 |
| oi100-ttd-default | 36968 | 0.4 | 0.48 | 0.46 | 11499 | 0.08 |

We explore synset matching as well; we match on noun synsets only, with the NLTK toolkit. We include all synonym nouns, hyponyms, hypernyms, alsosees and similartos, a broad matching strategy whose refinement we leave to future work. Synset matching also results in much slower subset matching, and also results in very small improvements; therefore, we use simple subset matching for the majority of our experiments.

### E.4 PERFORMANCE COMPARISON OF SUBSET MATCHING STRATEGIES ON YFCC AND LAION

We train baseline 2M and 4M models on random subsamples of YFCC. We then use subset matching strategies as described in Section E. We note that in1k-ours is more robust than in1k-default matching on YFCC, controlling for size.

We also try the opposite strategy – targeting the 10M samples in YFCC which were NOT matched. Even under this rather punishing transformation, we find that a VL model is able to learn a low-level representation of ImageNet (around 12 percent accuracy and an average of 8 percent under shift, around one third of a baseline YFCC model). The performance of a subset matched model under these circumstances would be zero, because it would not have any samples to train on.

**Does subset matching favor certain datasets?**

Subset matching strategies only reach parity with VL models when we approximately control for dataset utilization (the number of samples the model evaluates), indicating that this factor is important to consider when attempting to answer this question. Could it be that subset matching underperforms on non-robust datasets because it simply matches fewer samples?

This turns out not to be the case; we find that on YFCC and LAION, a subset match is found around 20 percent of the time, and on CC12M, around 25 percent of the time, resulting in roughly similar dataset utilization.

**Single-class subset matching usually outperforms other filtering strategies**

As shown earlier, subset matching strategy can impact both distributional robustness and accuracy during training, and changes do not necessarily affect both measures in equal proportion.

We consider a range of variations on simple subset matching to see if they offer any improvement, such as fuzzy matching using Levenshtein distance and synset matching, but in our experiments, there were few if any advantages to these techniques.

We also experiment with changing the term-matching dictionary and the matching strategy; for more information on this, please refer to Table E.2.

Overall, we find that single-class, non-strict matching on a relatively limited set of terms provides the best balance of accuracy, speed and dataset utilization, and that all subset matching strategies perform worse than ground-truth labels, even when controlling for dataset size.

**Prefiltering samples improves caption quality**

We find in Table E.2 that the caption noise profiles of ImageNet-100 and OpenImages-100 differ dramatically.

- On ImageNet-100, simple subset matching on flickr-captions achieves high dataset utilization and accuracy

- On OpenImages-100, the same strategy shows lower accuracy and much lower dataset utilization

Since both datasets use the same caption source and the same image source, data alone is unlikely to explain this discrepancy.

Instead, we suspect the difference exists because of the strategies used to assemble these datasets.

OpenImages labelers applied labels to images which had not been selected with any particular objective in mind. Kuznetsova et al. (2020) ImageNet labelers applied labels to images which had been preselected with the intent of building a class-balanced dataset for 1000 classes chosen in advance. Deng et al. (2009)

We cannot verify the accuracy of subset-matching on YFCC-100 or LAION-100 because there are no human labels for these datasets. However, we know that LAION samples used a much stronger prefiltering strategy than YFCC. Schuhmann et al. (2021b); Thomee et al. (2016) Therefore, it is possible that subset matching's strong performance on LAION can be attributed to this difference alone.

**Subset matching hit rate is a good signal for subset matching accuracy**

On OpenImages-100, we observed that as raw match count decrease dramatically, the accuracy of matched samples also decreased when matches took place. Table E.2 This fact offers one possible guideline for when subset matching is better than VL; it performs better when there are a relatively high number of subset matches, possibly because hit rate correlates with accuracy.

The hit rate on YFCC-100 is around 2 percent, whereas the hit rate on LAION-100 is around 3.5 percent. This difference is certainly substantial enough that it could be a signal that subset matching will be a more successful approach.

**Machine labels complement subset-matched labels**

When classes are known in advance, machine labeling can be a good strategy for learning unsupervised image data. One advantage of machine labeling is that if labels are accurate, dataset utilization has the potential to be much higher.

In Table 5, we find that on ImageNet-100, 90 percent of the labels from a CLIP ViT-L model match the ground truth label. This is very comparable to that model's zero-shot validation accuracy on ImageNet-val-subset. However, when we train a CE model on the CLIP labels, we find the machine labels outperform subset matching with size control, performing nearly as well as ground-truth labels.

The most likely explanation for this somewhat puzzling result is that the distribution of CLIP error is 'helpful' to model learning – results in this vein have been shown by Goh et al. (2021). The distribution of error in subset matched models, as we note in Section 2, closely approximates random noise.

Why might CLIP error be more helpful? One possible explanation for this difference is that VL-loss models always have available a very large space of potential classes for matching. Consider examples like "lioness" for the class "lion", or "lamp shade" for "lampshade" – subset matching approaches miss these positive matches (unless we use heuristic methods to correct them), but in a VL model, the most predictive token would likely remain the same. Since we know from Section G that bag-of-words contrastive loss VL models are not sensitive to token position in the string, this could result in a correct classification where a subset-matching model would fail.

Machine labeling can also be useful for estimating the accuracy of subset matched labels. On LAION-100, we find that 60 percent of labels are in agreement between subset matching and labels from a CLIP ViT-L model.

We leave to future work the question of how models with a wider range of predefined classes would perform on such a task.

Overall, we find that machine labeling and caption supervision serve complementary roles; machine labeling does not directly rely on captions, but can still benefit from them. In particular, subset matching can prefilter incoming image data to target classes of interest.

**The use cases for subset matching** We find that simple subset matching is a powerful technique for learning on unsupervised image / caption data. We make the following recommendations for applying this technique;

1. Both flickr-style tags and descriptions and alt-text can be effective when used for subset matching
2. Subset matching techniques are much more effective when image-caption data has been roughly filtered or supervised, even if it has not actually been labeled; for instance, all images that match terms in a search engine, or all images that maximize the dot product of a clip model.
3. Subset matching relies on terms which are relatively common, and so works best with objects with unique names that happen to often be the subject of an image; English cocker spaniel, for instance
4. If the terms are uncommon or difficult to nail down, but the captions are still expected to contain them, then a VL model may perform better
5. When data is unfiltered, the hit rate of subset matching is a good barometer for how accurate the matches will be
6. If hit rate is low, so is accuracy, and VL will probably perform better than subset matching
7. Augmenting or ensembling with machine labels works well in conjunction with captioning; captions generated from machine labels can provide additional signal for VL and subset matching when captions are noisy

## F    DETAILS ON CAPTIONNET

The most important new contribution in CaptionNet is ImageNet-100. To the best of our knowledge, ImageNet-100 is the only version of ImageNet which duplicates the original distribution's class balance and supervision properties (ImageNet is not perfectly class balanced, but it does not contain any long-tail classes; all classes in ImageNet have at least 750 samples), while also being fully captioned with original web-scraped labels. We find that both VL and CE models trained on relatively small amounts of data can achieve high base accuracy on some CaptionNet subsets, making it possible for the first time to compare model distributional robustness while controlling for base accuracy.

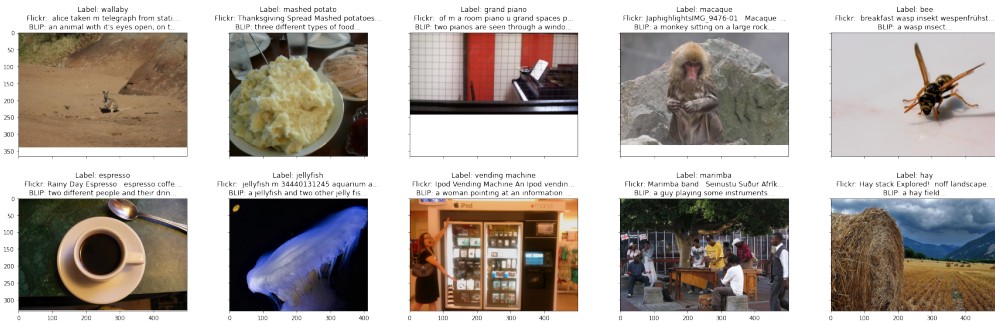

Figure 7: **ImageNet-100 samples from CaptionNet.**

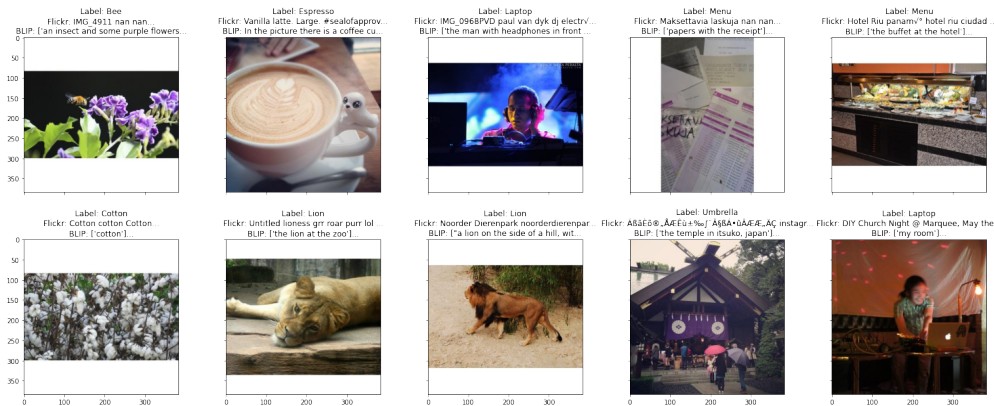

Figure 8: **OpenImages-100 samples from CaptionNet.**

In Table 10, we discuss in detail the supervision strategy used for CaptionNet, with a per-class breakdown of each class.

An overview of the supervision process follows;

- All samples were supervised by the authors of the paper

- Samples were sourced from flickr using the available API, sorted by 'interesting', with safesearch enabled, searching only samples with Creative Commons licenses

- Additional filtering terms were passed to the API in order to eliminate commonly encountered confounds in the search terms

- After the search term was selected, items were downloaded in bulk

- All downloaded samples were then individually tagged by the researchers as either "in-class" or "out-of-class", using reference photographs from each class as a baseline comparison

We found that classes varied widely along several vectors;

- Some classes had far greater availability than others (ranging from 450,000 to 283 available samples)

- Some classes were much cleaner than others (ranging from 100 percent clean to around 25 percent)

- Some classes tended to be the 'subject' of photographs, such as dog breed, while others, such as mashed potato, tended to be featured as secondary items in the background of a photograph of something else

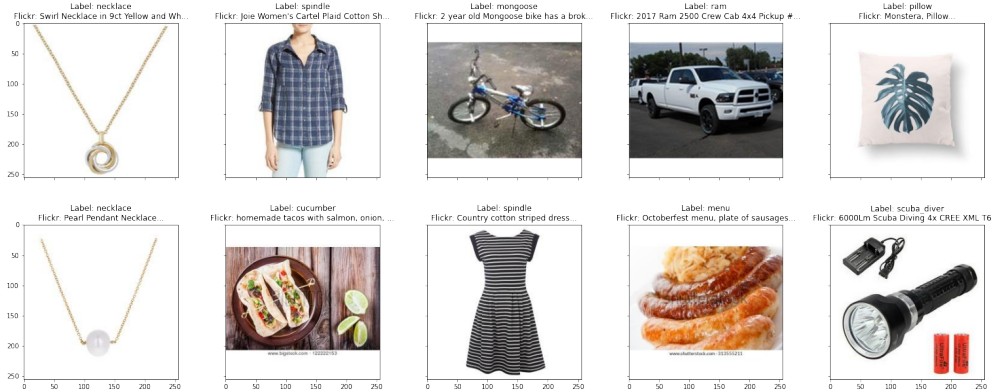

Figure 9: **LAION-100 samples from CaptionNet.**

### F.1 DATASET CONSTRUCTION

The 100 classes in CaptionNet were selected randomly from a subset of all classes with more than 600 captions available in ImageNet-Captions (Fang et al., 2022). The list of classes selected is available in Section I. We note that this approach introduces a potential bias in class selection, since it may be that captions were still available for those images ten years after ImageNet was originally constructed for some reason that correlates with properties we are interested in studying; however, we feel that the risk of this is outweighed by the many benefits of having such a dataset available for study.

Table 10: **CaptionNet Supervision: Search Terms and Sample Quality**
*Since many of the findings in our paper highlight the importance of both the amount and type of label noise, this table records statistics pertaining to our filtration process for the new samples in in100. In the search term field, a - symbol indicates that all samples which included that word in the title, tags or description were NOT matched. Boolean OR, AND, and "" symbols behave as they typically do.*

| in1k classname | search term | good samples | total samples | avail. samples | pct. good |
|---|---|---|---|---|---|
| lion | lion | 962 | 1000 | 450000 | 0.96 |
| wine bottle | wine bottle | 925 | 1000 | 29500 | 0.93 |
| book shop | bookstore | 816 | 984 | 83000 | 0.83 |
| parking meter | parking meter | 377 | 1000 | 9500 | 0.38 |
| african elephant | african elephant | 885 | 1000 | 44000 | 0.89 |
| bagel | bagel | 699 | 988 | 20500 | 0.71 |
| tarantula | tarantula | 667 | 981 | 9000 | 0.68 |
| ice cream | ice cream | 741 | 984 | 154500 | 0.75 |
| fig | fig | 517 | 1000 | 46000 | 0.52 |
| shoe shop | shopping shoes | 425 | 1000 | 13000 | 0.43 |
| french bulldog | french bulldog | 887 | 996 | 7500 | 0.89 |
| hen | hen | 412 | 1000 | 73000 | 0.41 |
| guacamole | guacamole | 683 | 998 | 6500 | 0.68 |
| broccoli | broccoli | 679 | 997 | 19000 | 0.68 |
| howler monkey | howler monkey | 817 | 847 | 9000 | 0.96 |
| scuba diver | scuba diver | 827 | 1000 | 15000 | 0.83 |
| spindle | "spindle wool, spindle -wool thread" | 311 | 867 | 867 | 0.36 |
| lhasa | lhasa dog | 719 | 1000 | 2500 | 0.72 |
| traffic light | stoplight | 622 | 991 | 5500 | 0.63 |
| lionfish | lionfish | 552 | 897 | 6500 | 0.62 |

| | | | | | |
|---|---|---|---|---|---|
| popsicle | popsicle -animal -sticks -animals -insect -insects -icicle -garden -sticks -icicles -gardens -toes -label -labels | 638 | 943 | 7500 | 0.68 |
| lampshade | lampshade | 446 | 807 | 6500 | 0.55 |
| spiderweb | spiderweb -spiderman -halloween -pumpkin -butterfly -pleiades -nebula -stars | 832 | 996 | 17500 | 0.84 |
| lifeboat | lifeboat | 572 | 1000 | 13000 | 0.57 |
| cucumber | cucumber -sea -spider -beetle -flower -spiral | 730 | 999 | 26500 | 0.73 |
| english springer | english springer spaniel | 772 | 993 | 3500 | 0.78 |
| macaw | macaw | 972 | 1000 | 13500 | 0.97 |
| mailbox | mailbox | 900 | 1000 | 36500 | 0.9 |
| peacock | peacock -butterfly | 966 | 999 | 72000 | 0.97 |
| bee | bumblebee OR wasp OR hornet -jet -airplane -helicopter -navy -aircraft -comic -RIAT -military -Helicopter -Helicopters -helicopters -aviation -Hudson -car -basketball -sports -Transformers -cosplay -disfrazado -costume -transformer AND flower | 686 | 761 | 110000 | 0.9 |
| dungeness crab | dungeness AND crab -restaurant -breakfast -lunch -dinner -shack -creels -traps -cannery | 474 | 1000 | 1500 | 0.47 |
| banana | banana -plant -blossom -flower -seed -seedlings -tree -spider -leaf -abstract -bay -band -festival -doll -sexy -sexiest -bread -soup -puree -smoothie -car -plantation -cake -cream -monkey -pudding -zoo -republic -boxes -buying -selling -vendor -bridge -scone -moon | 793 | 995 | 65000 | 0.8 |
| corn | corncob | 354 | 1000 | 1000 | 0.35 |
| lemon | lemon -plant -blossom -flower -seed -seedlings -tree -spider -leaf -abstract -bay -band -festival -doll -sexy -sexiest -bread -soup -puree -smoothie -car -plantation -scent -fresh -cleaner -butterfly -grove -shots -car -sunrise -paint -graffiti -origami -cake -cream -pudding -boxes -buying -selling -vendor -bridge -scone -don -lime | 693 | 1000 | 65000 | 0.69 |
| marimba | marimba instrument | 127 | 283 | 283 | 0.45 |

| orange | orange food fruit -plant -blossom -flower -seed -seedlings -tree -spider -leaf -abstract -bay -band -festival -doll -sexy -sexiest -bread -soup -puree -smoothie -car -plantation -cake -cream -monkey -pudding -zoo -republic -boxes -buying -selling -vendor -bridge -scone -moon -cupcake -cake -sales -seller -pancakes -crepes -crep -crepe -pancake -cookie -flavored -juice -soda -pop -beach -island -cove -grove -street -drive -tea -curd -marmalade -bars -cabs -chicken -cheesecake -pie -milk | 744 | 1000 | 4000 | 0.74 |
|---|---|---|---|---|---|
| bell pepper | bell pepper vegetable -plant -blossom -flower -seed -seedlings -tree -spider -leaf -abstract -bay -band -festival -doll -sexy -sexiest -bread -soup -puree -smoothie -car -plantation -cake -cream -monkey -pudding -zoo -republic -boxes -buying -selling -vendor -bridge -scone -moon -cupcake -cake -sales -seller -pancakes -crepes -crep -crepe -pancake -cookie -flavored -juice -soda -pop -beach -island -cove -grove -street -drive -tea -curd -marmalade -bars -cabs -chicken -cheesecake -pie -milk -market -spice | 392 | 505 | 505 | 0.78 |
| espresso | espresso coffee -maker -machine -beans -building -exterior -window | 828 | 1000 | 22000 | 0.83 |
| mashed potato | mashed potato | 635 | 996 | 10000 | 0.64 |
| stingray | stingray water -dolphin -shark -cruise -boat -scuba -fish | 600 | 983 | 2000 | 0.61 |
| flagpole | flagpole -lighthouse -church -bank -station | 614 | 991 | 7000 | 0.62 |
| teapot | teapot -tea -flower -tower -building -dome -art -fashion -vase -store -stores -shop -shops -Sagittarius -project365 -fountain -candle -mug -teacup -keg -vessel -amphora -urn -coffeepot | 660 | 997 | 10500 | 0.66 |
| umbrella | umbrella | 911 | 1000 | 126000 | 0.91 |
| beer bottle | beer bottle -house -door -brewery -glass -cap | 909 | 1003 | 19000 | 0.91 |
| barn | barn -swallow -owl -bird | 980 | 1000 | 115000 | 0.98 |
| christmas stocking | christmas stocking fireplace | 317 | 779 | 779 | 0.41 |
| magpie | magpie -screenshots -moth -butterfly -coprinopsis -thieving -mushroom | 736 | 983 | 25500 | 0.75 |
| mitten | mitten glove | 800 | 995 | 1500 | 0.8 |

| ram | ram sheep -Church -window -Window -church -school -dance -parade -festival -celebration -festivities -community -fair -ewe -fox -lamb -bird -cat -dog -Dodge | 742 | 1000 | 3000 | 0.74 |
|---|---|---|---|---|---|
| warthog | warthog animal -zebra -cheetah -leopard -giraffe -gazelle -hippo -rhino -donkey -armadillo -elephant -crocodile -lion -leopard -impala -cat -monkey -bird | 946 | 997 | 2500 | 0.95 |
| goose | geese | 474 | 500 | 69000 | 0.95 |
| bubble | soap bubble -dancer -dance -fairy -tree -leaf -leaves -flowers -water -toy -art -abstract -museum -dog -cat -butterfly -food -wine -beer -chocolate -Chocolate | 414 | 500 | 5000 | 0.83 |
| cougar | cougar animal -warthog -mascot -zebra -cheetah -leopard -giraffe -gazelle -hippo -rhino -donkey -armadillo -elephant -crocodile -lion -leopard -impala -cat -monkey -bird -lake -Lake -river -River -blonde -Blonde -woman -girl -milf -bear -cliff -Cliffs -cliffs -military -wallaby -horse -jet -print | 297 | 500 | 1000 | 0.59 |
| daisy | daisy flower | 500 | 500 | 52000 | 1 |
| menu | menu | 431 | 500 | 92000 | 0.86 |
| bald eagle | bald eagle | 475 | 500 | 33500 | 0.95 |
| necklace | necklace jewelry -brooch -pendant -creation -earring -earrings -bracele -ring -Engraver -bauble -anklet | 478 | 500 | 12500 | 0.96 |
| chickadee | chickadee bird -Goldfinch -goldfinch -robin -thrush -jay -cardinal -woodpecker -wren -hawk -raven -titmouse -nuthatch | 494 | 500 | 9000 | 0.99 |
| stone wall | ”””stone wall””” | 424 | 500 | 32000 | 0.85 |
| flamingo | flamingo bird | 476 | 500 | 38500 | 0.95 |
| gas pump | gas station | 348 | 500 | 41000 | 0.7 |
| vulture | vulture bird -hawk -crow -eagle | 489 | 500 | 15500 | 0.98 |
| pizza | ”””pizza pie”” -Fest -festival -summit -experience -party -band -moon -parade -Parade -harvard -mosaic -montage” | 305 | 500 | 1000 | 0.61 |
| wallaby | wallaby -warthog -mascot -zebra -cheetah -leopard -giraffe -gazelle -hippo -rhino -donkey -armadillo -elephant -crocodile -lion -leopard -impala -cat -monkey -bird -koala -sports -kangaroo -soccer -football -food -church -hills -stadium -tribute -grass -rugby -apartment -car | 369 | 500 | 10000 | 0.74 |
| hay | haystack field -hole -trail -poster -sign | 360 | 500 | 1000 | 0.72 |
| grand piano | ”kawai grand piano, steinway grand piano” | 312 | 455 | 455 | 0.69 |
| laptop | laptop | 443 | 500 | 98000 | 0.89 |
| dishwasher | dishwasher appliance | 191 | 268 | 268 | 0.71 |

| cricket | cricket -batting -sports -team -match | 337 | 500 | 44000 | 0.67 |
|---|---|---|---|---|---|
| sea slug | nudibranch | 468 | 500 | 12500 | 0.94 |
| mongoose | mongoose -bike -bicycle -park -tree -joe -rocket -military -airplane -toy -car | 379 | 500 | 5000 | 0.76 |
| siamese cat | siamese cat -bangkok -flower -snake -campaign -wat -costume -cosplay -festival | 416 | 500 | 13000 | 0.83 |
| freight car | freight car | 491 | 500 | 70500 | 0.98 |
| vending machine | ”””vending machine””” | 411 | 500 | 13000 | 0.82 |
| bottlecap | bottlecap -tab | 448 | 500 | 3500 | 0.9 |
| acorn | acorn -woodpecker -fairy -squirrel -weevil -travel -squash -street | 352 | 500 | 25000 | 0.7 |
| feather boa | feather boa | 135 | 500 | 2000 | 0.27 |
| macaque | macaque | 485 | 500 | 14500 | 0.97 |
| bolete | boletus | 444 | 500 | 3500 | 0.89 |
| border terrier | ”””border terrier””” | 422 | 500 | 1500 | 0.84 |
| barbell | barbells | 352 | 500 | 1000 | 0.7 |
| fly | housefly | 398 | 500 | 1500 | 0.8 |
| suspension bridge | suspension bridge | 432 | 500 | 33500 | 0.86 |
| jellyfish | jellyfish | 477 | 500 | 46500 | 0.95 |
| barbershop | barbershop -quartet -singers | 430 | 500 | 9000 | 0.86 |
| koala | koala | 458 | 500 | 32500 | 0.92 |
| bannister | bannister staircase | 174 | 183 | 183 | 0.95 |
| pillow | pillow -talk -fight -cat -dog -moss -sky -cloud -sky | 420 | 500 | 34500 | 0.84 |
| bib | baby bib -shower -food | 406 | 500 | 1500 | 0.81 |
| junco | junco bird -finch -sparrow -thrush -cardinal -woodpecker -jay | 475 | 500 | 7000 | 0.95 |
| chainlink fence | chainlink fence | 375 | 500 | 3500 | 0.75 |
| soccer ball | ”””soccer ball”” -match -game -milky -beach -Lewes” | 349 | 500 | 2500 | 0.7 |
| stupa | stupa | 418 | 500 | 23500 | 0.84 |
| quail | quail bird -finch -sparrow -thrush -cardinal -woodpecker -jay -partridge -rabbit -hawk -avocet -deer -dog -wolf -coyote -gopher -eagle -vole -molerat -butterfly | 396 | 500 | 11000 | 0.79 |
| padlock | padlock | 378 | 500 | 9500 | 0.76 |
| great white shark | ”””great white shark””” | 309 | 500 | 2000 | 0.62 |
| totem pole | ”””totem pole”” wood” | 383 | 500 | 1000 | 0.77 |
| ant | ant insect | 447 | 500 | 18000 | 0.89 |
| bison | bison | 429 | 500 | 41500 | 0.86 |
| greenhouse | greenhouse | 407 | 500 | 82000 | 0.81 |

**Adding BLIP Captions to CaptionNet**

Since we could not find human-authored captions for ImageNet, we used BLIP Li et al. (2022) to generate descriptive captions on ImageNet-100. BLIP often uses word fragments to describe objects, so we used a spell checker as a simple intervention to improve the quality of BLIP captions. Finally, because BLIP's vocabulary does not include many of the specialized classes available in ImageNet, we augmented the BLIP captions with Flickr image titles, the form of text which is most commonly available for an image. We used top p=0.9, max length=40, min length=5, repetition penalty=1.1.

We repeated the process for OpenImages-100. However, we used human-authored captions sourced from Pont-Tuset et al. (2020) instead of BLIP whenever available; around 16,000 out of the 135,000 OpenImages-100 samples had human-authored captions.

Table 11: **Captioning strategy can affect model distributional robustness and accuracy.** *VL models perform better on OpenImages when flickr-captions are replaced with synthetic captions (BLIP+Title captions), but the same captioning method provides no benefits on ImageNet-100.*

| Model | Technique | Val. Acc. (in100) | Avg. Rob. | E.R.R. |
|---|---|---|---|---|
| ImageNet-100 BLIP-Caption | VL | $0.574 \pm 0.014$ | $0.216 \pm 0.008$ | 0.376 |
| ImageNet-100 Flickr-Caption | VL | $0.574 \pm 0.014$ | $0.217 \pm 0.008$ | 0.378 |
| OpenImages-100 BLIP+Human-Caption | VL | $0.283 \pm 0.012$ | $0.131 \pm 0.008$ | 0.464 |
| OpenImages-100 Flickr+Human-Caption | VL | $0.225 \pm 0.012$ | $0.11 \pm 0.008$ | 0.489 |
| OpenImages-100 Flickr-Caption | VL | $0.197 \pm 0.012$ | $0.095 \pm 0.008$ | 0.482 |

## G  CLIP DOES NOT LEARN A LANGUAGE MODEL

We show that, evaluated by the CE definition of a language model in the literature, as well as by our expectations as human beings, CLIP's text tower does not learn a language model.

Fang et al. (2022) noted that on most natural distribution shifts, models trained with language information from a captioned subset of ImageNet follow the same trend as models trained without it, with neither coming close to the distributional robustness of VL models.

We conducted a range of experiments in Table G to verify this.

1. Zeroshot and fully trained scrambling: we randomly scrambled the word order of captions and trained a model for the full training duration. We saw only a 1% loss in zero-shot accuracy when the model was trained in this fashion, and no loss at all when we scrambled the word order of the prompts used to generate the text embeddings. This finding shows that despite the existence of a positional encoder, CLIP's text tower is invariant w.r.t. word position on its input captions.

2. We train a "simple captions" model of yfcc, with "An image of a CLASSNAME", where the class name is all the wordnet-recognized nouns and adjectives in the captions, and see very little change in effective robustness. This indicates that the model pays very little attention to verbs, adverbs and non-wordnet tokens, at least for the purposes of ImageNet zero-shot accuracy.

3. We train a "simpler captions" model on YFCC, captioned "An image of a CLASSNAME", where the class name is a single version of an ImageNet-1k classname. We find that accuracy decreases, but even under this highly destructive transformation, effective robustness remains on the line.

4. We test prompt ensembling by eliminating all but one prompt during inference when computing CLIP's predictions. We try scrambling this prompt and leaving it unscrambled. We find very little change in accuracy and none in effective robustness. This indicates that prompt selection and prompting methods may improve accuracy, but do not affect effective robustness.

5. We fully trained a language model with a shift cipher applied to all of the letters in the captions. While this has a significant effect on accuracy (a drop of 8%), effective robustness holds nearly constant. This shows that CLIP's tokenizer aids accuracy but offers no effective robustness benefit (since a shift cipher destroys nearly all word-wise tokenization and forces the model to tokenize one letter at a time) and that the model is not more robust because of some special understanding of the text data itself, such as learning the frequency distribution of letters or tokens as a way of guessing captions.

Throughout our experiments, we find that interventions that affect the integrity of the caption space do have impacts on the overall accuracy of the model, they do not change the effective robustness

trend line, indicating that the changes in effective robustness cannot be attributed to the properties of natural language which we ablate, such as the tokenizer, sentence-wise attention mechanisms, prompt ensembling or even class-independent caption content.Table G

| name | dataset | in-val | in-a | in-r | in-s | in-v2 | Avg. Rob. | E.R.R. |
|---|---|---|---|---|---|---|---|---|
| Baseline | yfcc15m | 0.324 | 0.136 | 0.2232 | 0.073 | 0.2804 | 0.178 | 0.55 |
| Simple Captions | yfcc15m | 0.31 | 0.1576 | 0.21 | 0.06 | 0.2737 | 0.175 | 0.566 |
| Simple Captions & Filtering | yfcc15m | 0.101 | 0.038 | 0.0736 | 0.008 | 0.0898 | 0.052 | 0.521 |
| Simpler Captions & Filtering | yfcc15m | 0.09 | 0.0441 | 0.0694 | 0.011 | 0.0804 | 0.051 | 0.573 |
| Shift Cipher | yfcc15m | 0.238 | 0.139 | 0.1657 | 0.049 | 0.2016 | 0.139 | 0.584 |
| No Ensembling | yfcc15m | 0.284 | 0.131 | 0.188 | 0.057 | 0.2394 | 0.154 | 0.541 |
| Token Order Scrambled | yfcc15m | 0.333 | 0.16 | 0.237 | 0.069 | 0.295 | 0.19 | 0.571 |

## G.1 CAPTIONS IN DATASETS ARE OFTEN SIMILAR

In Table G.1, we observe that by many common measures, alt-text and flickr captions are quite similar; whatever explanations may exist for the performance differences, it is not easily summarized by these measures.

Table 12: **Alt-text and flickr captions do not differ substantially by many measures.** *We observe that by many measures, alt-text captions, are very similar to flickr captions, making it difficult to determine why models trained on alt-text captions tend to be more robust.*

| Dataset | Size | Non-english Capt. Freq. | Avg. Capt. Len. | Std. Dev. Capt. Len. | Num. Uniq. Tokens | Supervision Strat. | Filtering Strat. |
|---|---|---|---|---|---|---|---|
| LAION-15m (alt-text) | 13775512 | 2300223 | 59.79 | 70.43 | 6020847 | unsupervised | filtered |
| YFCC-15m (flickr) | 14825134 | 1367899 | 114.98 | 77.36 | 6747440 | unsupervised | unfiltered |
| OpenImages-100 (flickr + annot) | 136175 | 53346 | 164.21 | 328.06 | 469154 | supervised | unfiltered |
| ImageNet-100 (flickr) | 124351 | 23563 | 195.34 | 274.17 | 355452 | supervised | filtered |
| YFCC-100 (flickr) | 374975 | 16941 | 181.14 | 249.24 | 480458 | unsupervised | unfiltered |
| LAION-100 (alt-text) | 453860 | 69189 | 65.32 | 64.05 | 402230 | unsupervised | filtered |

## H  ALTERNATE TRAINING SCHEMES

One fairly immediate explanation for the distributional robustness of VL models would be that we are witnessing a kind of overfitting which happens whenever a model is fine-tuned on a training dataset. If this is the case, then if we had a method for reverting the overfitting, we would be able to shift models to a different line with respect to effective distributional robustness.

In Fig. H, we explore this possibility, focusing on two alternative training schemes in particular, Wise-FT and LiT-Tuning.

Radford et al. (2021) ran fine-tuning experiments on certain datasets and found that robustness declined as base accuracy increased, indicating that fine-tuning makes CLIP models (somewhat) less robust, fitting a middle line in-between VL and CE models.

Wortsman et al. (2022) ran a series of experiments interpolating the weights of zero-shot CLIP with its fine-tuned counterparts and showed that for certain distribution shifts, it is possible to find a 'sweet spot' where both i.d. and o.o.d. accuracy increase. We test the wise-ft method on ViT-L and find that under this intervention, distributional robustness does not scale proportionately with accuracy, instead holding essentially constant as base accuracy increases.

We also evaluated the pre-trained LiT-tuned models released by Zhai et al, and trained several LiT-tuned models ourselves in order to compare their performance to fully-trained VL models, extending the results in Beyer et al. Our findings are as follows –

1. Like Wise-FT, LiT-tuning produces models whose i.d. / o.o.d accuracy trade-off fits a line between that of traditional models and VL models – more robust than the former, less robust than the latter. The only exception we found was when we LiT-tuned the vision tower of a ViT trained on the CLIP objective – in this case, LiT-tuning decreased base accuracy while holding effective robustness constant (the near-opposite effect of Wise-FT)

2. LiT-tuning offers negative benefit for fully trained VL models, suggesting that it can only hope to approach, rather than exceed, the accuracy of its baselines

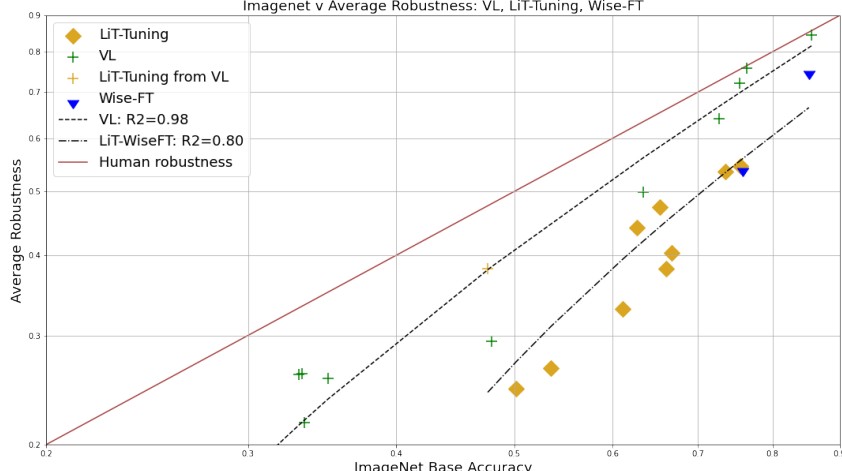

Figure 10: **Wise-FT, optimized to balance id/ood accuracy, fits the LiT-tuned effective robustness line.**

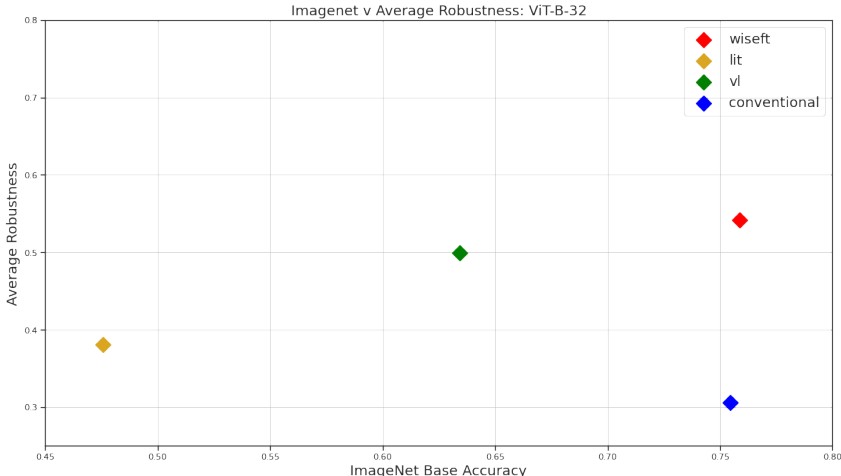

Figure 11: **LiT-tuning on a VL-trained image tower reduces accuracy without altering effective robustness, suggesting that VL pretraining is at least as robust as LiT-tuning.***Wise-FT tuning greatly increases base accuracy and slightly improves effective robustness, at the cost of zero-shot capability. CE from-scratch training matches Wise-FT accuracy, but sacrifices effective robustness and zero-shot.*

3. LiT-tuning performance tends to closely correlate to the base accuracy of the underlying vision model

4. Intriguingly, we find that this is true regardless of the specific dataset used for LiT-tuning – LiT-tuned models trained on small amounts of data are able to recover accuracy on out-of-distribution tasks even when very little data from that distribution shift appears in the pretraining data

5. These experiments suggest that some degree of effective robustness is "locked away" in many vision models, but is lost during the training process, but that certain techniques are able to increase effective robustness disproportionate to the loss in base accuracy, pushing the model 'above the line' we would normally expect. Furthermore, if the distribution shift of interest is known and well-defined, it is possible to select a tuning to optimize for that shift

Taken together, we can conclude that effective robustness cannot be explained by overfitting alone – if it were, then we would expect interpolation-type interventions to be capable of lifting models to the effective robustness line, which they do not.

# I    CLASS FREQUENCY COUNTS FOR IN100 SUPERVISED DISTRIBUTIONS

## I.1    IMAGENET-100

'african bush elephant': 2082, 'mailbox': 1846, 'peafowl': 1844, 'macaw': 1689, 'barn': 1689, 'wine bottle': 1648, 'lion': 1643, 'umbrella': 1608, 'mitten': 1595, 'warthog': 1565, 'french bulldog': 1550, 'spider web': 1544, 'beer bottle': 1531, 'scuba diver': 1514, 'bookstore': 1493, 'ice cream': 1475, 'traffic light': 1468, 'koala': 1456, 'espresso': 1450, 'banana': 1444, 'magpie': 1439, 'bottle cap': 1437, 'howler monkey': 1428, 'orange': 1425, 'teapot': 1414, 'english springer spaniel': 1407, 'lhasa apso': 1404, 'bagel': 1401, 'tarantula': 1378, 'ram adult male sheep': 1377, 'lemon': 1370, 'cucumber': 1360, 'bee': 1355, 'popsicle': 1351, 'broccoli': 1339, 'guacamole': 1335, 'fig': 1323, 'menu': 1291, 'mashed potatoes': 1283, 'stingray': 1270, 'flagpole': 1268, 'chickadee': 1253, 'lionfish': 1221, 'goose': 1220, 'junco': 1216, 'daisy': 1198, 'freight car': 1196, 'lifeboat': 1194, 'hay': 1193, 'bubble': 1173, 'dungeness crab': 1168, 'sea slug': 1165, 'vulture': 1158, 'suspension bridge': 1152, 'laptop computer': 1146, 'chain link fence': 1142, 'bald eagle': 1142, 'flamingo': 1137, 'padlock': 1133, 'jellyfish': 1128, 'stupa': 1125, 'ant': 1119, 'parking meter': 1115, 'wallaby': 1114, 'fly': 1108, 'border terrier': 1107, 'greenhouse': 1104, 'necklace': 1100, 'macaque': 1095, 'stone wall': 1093, 'barbell': 1093, 'lampshade': 1087, 'siamese cat': 1085, 'barbershop': 1072, 'pillow': 1071, 'spindle': 1069, 'bolete': 1067, 'vending machine': 1066, 'totem pole': 1054, 'bison': 1049, 'quail': 1049, 'hen': 1045, 'shoe store': 1045, 'christmas stocking': 1043, 'baby bib': 1034, 'mongoose': 1026, 'bell pepper': 1022, 'cricket insect': 1018, 'acorn': 1015, 'gas pump': 1009, 'soccer ball': 1005, 'great white shark': 1002, 'pizza': 983, 'corn': 969, 'grand piano': 962, 'cougar': 920, 'marimba': 914, 'dishwasher': 810, 'baluster handrail': 804, 'feather boa': 759

## I.2    OPENIMAGES-100

'laptop computer': 55813, 'menu': 13159, 'grand piano': 10836, 'bee': 3688, 'umbrella': 3575, 'goose': 3477, 'wine bottle': 2318, 'necklace': 2224, 'traffic light': 1915, 'pizza': 1699, 'ice cream': 1618, 'hen': 1499, 'lion': 1300, 'pillow': 1205, 'banana': 1119, 'siamese cat': 1058, 'orange': 1035, 'lemon': 961, 'jellyfish': 797, 'stone wall': 718, 'baluster handrail': 697, 'broccoli': 614, 'cucumber': 609, 'greenhouse': 597, 'chain link fence': 594, 'cougar': 593, 'macaque': 588, 'vulture': 560, 'scuba diver': 559, 'beer bottle': 550, 'teapot': 549, 'french bulldog': 546, 'espresso': 528, 'soccer ball': 527, 'lampshade': 491, 'fly': 491, 'macaw': 488, 'peafowl': 487, 'stupa': 484, 'suspension bridge': 476, 'ant': 475, 'flamingo': 462, 'barn': 458, 'bell pepper': 458, 'padlock': 434, 'african bush elephant': 432, 'spider web': 421, 'feather boa': 416, 'koala': 415, 'mongoose': 409, 'bottle cap': 388, 'freight car': 387, 'vending machine': 370, 'bison': 363, 'bald eagle': 351, 'hay': 340, 'bubble': 339, 'daisy': 339, 'totem pole': 335, 'stingray': 333, 'chickadee': 322, 'mailbox': 315, 'barbell': 313, 'bagel': 308, 'bookstore': 307, 'gas pump': 303, 'wallaby': 300, 'marimba': 294, 'spindle': 290, 'bolete': 262, 'popsicle': 258, 'shoe store': 255, 'guacamole': 250, 'barbershop': 248, 'lionfish': 234, 'parking meter': 234, 'lhasa apso': 218, 'christmas stocking': 211, 'mashed potatoes': 193, 'great white shark': 186, 'magpie': 186, 'warthog': 170, 'sea slug': 166, 'cricket insect': 156, 'lifeboat': 141, 'baby bib': 138, 'junco': 138, 'dishwasher': 113, 'mitten': 101, 'acorn': 98, 'quail': 96, 'tarantula': 89, 'howler monkey': 66, 'english springer spaniel': 61, 'fig': 54, 'ram adult male sheep': 54, 'dungeness crab': 51, 'border terrier': 49

# J    CLASS FREQUENCY COUNTS FOR IN100 SUBSET MATCHING DISTRIBUTIONS, OPENAI LABELS, MC MATCHING

## J.1    IMAGENET-100

'orange': 1820, 'lion': 1788, 'barn': 1695, 'macaw': 1684, 'umbrella': 1583, 'banana': 1500, 'mitten': 1500, 'warthog': 1488, 'magpie': 1438, 'lemon': 1437, 'koala': 1435, 'espresso': 1400, 'bagel': 1376, 'howler monkey': 1337, 'tarantula': 1331, 'broccoli': 1299, 'fig': 1295, 'ice cream':

1285, 'cucumber': 1272, 'goose': 1231, 'daisy': 1224, 'junco': 1207, 'chickadee': 1193, 'teapot': 1175, 'french bulldog': 1166, 'vulture': 1150, 'stingray': 1142, 'guacamole': 1134, 'flamingo': 1126, 'lifeboat': 1120, 'ant': 1114, 'suspension bridge': 1109, 'greenhouse': 1100, 'lhasa apso': 1093, 'wallaby': 1073, 'stupa': 1073, 'bald eagle': 1063, 'lionfish': 1057, 'fly': 1055, 'english springer spaniel': 1051, 'necklace': 1048, 'bison': 1047, 'barbell': 1042, 'mailbox': 1041, 'quail': 1037, 'macaque': 1032, 'padlock': 1026, 'hen': 1024, 'pizza': 995, 'pillow': 995, 'acorn': 993, 'vending machine': 976, 'bottle cap': 969, 'stone wall': 968, 'popsicle': 955, 'spider web': 949, 'totem pole': 934, 'spindle': 920, 'bookstore': 903, 'bubble': 893, 'border terrier': 889, 'mongoose': 888, 'corn': 874, 'parking meter': 866, 'flagpole': 864, 'dungeness crab': 862, 'marimba': 862, 'peafowl': 848, 'bee': 840, 'bell pepper': 821, 'menu': 758, 'wine bottle': 734, 'great white shark': 733, 'jellyfish': 703, 'dishwasher': 701, 'soccer ball': 700, 'beer bottle': 663, 'grand piano': 600, 'bolete': 576, 'hay': 547, 'gas pump': 541, 'christmas stocking': 534, 'traffic light': 479, 'cougar': 471, 'scuba diver': 470, 'feather boa': 435, 'african bush elephant': 408, 'siamese cat': 358, 'lampshade': 352, 'barbershop': 349, 'baby bib': 258, 'freight car': 119, 'laptop computer': 46, 'sea slug': 37, 'shoe store': 32, 'cricket insect': 19, 'baluster handrail': 2

## J.2 YFCC-100

'grand piano': 62610, 'orange': 37182, 'fly': 30889, 'lion': 16043, 'bee': 14164, 'pizza': 12084, 'barn': 11854, 'goose': 11200, 'ice cream': 10556, 'greenhouse': 9479, 'menu': 7463, 'umbrella': 7164, 'banana': 6933, 'bubble': 6838, 'corn': 6835, 'cougar': 6619, 'lemon': 6439, 'daisy': 5386, 'scuba diver': 5044, 'cricket insect': 4702, 'laptop computer': 4601, 'ant': 4465, 'hay': 4427, 'peafowl': 4140, 'pillow': 4137, 'flamingo': 3735, 'bookstore': 3668, 'necklace': 3201, 'bald eagle': 2912, 'ram adult male sheep': 2617, 'jellyfish': 2482, 'vulture': 2462, 'suspension bridge': 2439, 'espresso': 2189, 'mailbox': 2125, 'bison': 2073, 'flagpole': 2012, 'fig': 1973, 'hen': 1896, 'cucumber': 1815, 'bagel': 1746, 'koala': 1592, 'magpie': 1366, 'stone wall': 1337, 'spider web': 1296, 'acorn': 1277, 'popsicle': 1226, 'baluster handrail': 1182, 'vending machine': 1118, 'broccoli': 1114, 'junco': 1113, 'quail': 1108, 'stupa': 1043, 'feather boa': 1018, 'stingray': 971, 'macaw': 961, 'wallaby': 942, 'sea slug': 832, 'chickadee': 783, 'lifeboat': 781, 'baby bib': 774, 'mitten': 748, 'teapot': 728, 'macaque': 661, 'traffic light': 638, 'mashed potatoes': 625, 'african bush elephant': 600, 'tarantula': 593, 'barbershop': 537, 'gas pump': 520, 'padlock': 517, 'beer bottle': 433, 'warthog': 430, 'mongoose': 407, 'siamese cat': 395, 'guacamole': 393, 'parking meter': 381, 'spindle': 379, 'wine bottle': 370, 'dishwasher': 361, 'lampshade': 358, 'lhasa apso': 356, 'howler monkey': 314, 'lionfish': 296, 'shoe store': 285, 'soccer ball': 260, 'marimba': 168, 'freight car': 114, 'great white shark': 104, 'christmas stocking': 100, 'dungeness crab': 97, 'french bulldog': 94, 'bottle cap': 85, 'bolete': 80, 'chain link fence': 51, 'barbell': 23, 'english springer spaniel': 22, 'border terrier': 19

## J.3 LAION-100

'spindle': 68186, 'necklace': 59079, 'orange': 52221, 'pillow': 41020, 'laptop computer': 23319, 'lion': 14246, 'lemon': 12769, 'ram adult male sheep': 11550, 'bubble': 11079, 'barn': 9909, 'pizza': 9597, 'daisy': 8331, 'umbrella': 8323, 'banana': 7690, 'corn': 7140, 'menu': 6788, 'cougar': 6714, 'ice cream': 6539, 'cricket insect': 5696, 'peafowl': 4662, 'espresso': 4227, 'flamingo': 4029, 'goose': 3532, 'soccer ball': 3532, 'barbershop': 2963, 'dishwasher': 2853, 'bald eagle': 2678, 'fig': 2635, 'greenhouse': 2460, 'broccoli': 2348, 'teapot': 2298, 'acorn': 2164, 'cucumber': 2053, 'hay': 2023, 'wine bottle': 1824, 'scuba diver': 1818, 'bison': 1736, 'lampshade': 1497, 'mitten': 1457, 'french bulldog': 1435, 'stone wall': 1402, 'koala': 1394, 'bee': 1296, 'mailbox': 1199, 'padlock': 1126, 'stingray': 1115, 'bookstore': 1069, 'spider web': 976, 'macaw': 964, 'barbell': 913, 'christmas stocking': 887, 'traffic light': 825, 'vending machine': 808, 'popsicle': 780, 'quail': 768, 'chickadee': 744, 'bagel': 714, 'baluster handrail': 713, 'jellyfish': 706, 'bottle cap': 648, 'beer bottle': 603, 'flagpole': 589, 'bell pepper': 553, 'grand piano': 544, 'guacamole': 520, 'magpie': 481, 'suspension bridge': 477, 'african bush elephant': 459, 'baby bib': 451, 'wallaby': 423, 'stupa': 399, 'macaque': 350, 'gas pump': 335, 'great white shark': 333, 'mongoose': 308, 'junco': 302, 'siamese cat': 291, 'marimba': 289, 'hen': 272, 'tarantula': 257, 'lifeboat': 236, 'lionfish': 205, 'totem pole': 199, 'english springer spaniel': 192, 'warthog': 186, 'shoe store': 166, 'border terrier': 145, 'vulture': 118, 'feather boa': 116, 'lhasa apso': 105, 'sea slug': 90, 'howler monkey': 85, 'fly': 83, 'parking meter': 54, 'freight car': 50, 'ant': 44, 'dungeness crab': 36, 'chain link fence': 33, 'bolete': 14

## K    PER CLASS ACCURACY FOR SUBSET MATCHING, OPENAI CLASSNAMES, SC

### K.1    IMAGENET-100

'macaw': 0.81, 'barn': 0.9, 'umbrella': 0.85, 'lion': 0.92, 'mitten': 0.89, 'warthog': 0.9, 'magpie': 0.87, 'koala': 0.88, 'banana': 0.88, 'espresso': 0.89, 'bagel': 0.88, 'howler monkey': 0.87, 'tarantula': 0.87, 'orange': 0.86, 'lemon': 0.87, 'fig': 0.87, 'broccoli': 0.85, 'cucumber': 0.84, 'ice cream': 0.84, 'junco': 0.83, 'goose': 0.83, 'chickadee': 0.83, 'teapot': 0.82, 'daisy': 0.82, 'french bulldog': 0.82, 'vulture': 0.82, 'stingray': 0.81, 'guacamole': 0.81, 'flamingo': 0.81, 'lifeboat': 0.81, 'suspension bridge': 0.81, 'greenhouse': 0.8, 'lhasa apso': 0.81, 'ant': 0.8, 'stupa': 0.8, 'wallaby': 0.8, 'bald eagle': 0.8, 'lionfish': 0.82, 'english springer spaniel': 0.82, 'bison': 0.82, 'barbell': 0.82, 'macaque': 0.82, 'mailbox': 0.85, 'necklace': 0.84, 'quail': 0.84, 'padlock': 0.85, 'hen': 0.84, 'acorn': 0.83, 'pillow': 0.82, 'fly': 0.83, 'vending machine': 0.83, 'stone wall': 0.83, 'bottle cap': 0.83, 'popsicle': 0.83, 'spider web': 0.82, 'totem pole': 0.82, 'pizza': 0.82, 'spindle': 0.81, 'bookstore': 0.79, 'mongoose': 0.79, 'border terrier': 0.78, 'parking meter': 0.77, 'marimba': 0.77, 'flagpole': 0.77, 'dungeness crab': 0.78, 'peafowl': 0.77, 'bubble': 0.74, 'bell pepper': 0.74, 'bee': 0.7, 'corn': 0.68, 'menu': 0.68, 'great white shark': 0.67, 'wine bottle': 0.67, 'dishwasher': 0.65, 'soccer ball': 0.65, 'jellyfish': 0.65, 'beer bottle': 0.59, 'grand piano': 0.56, 'bolete': 0.55, 'gas pump': 0.52, 'christmas stocking': 0.51, 'hay': 0.48, 'traffic light': 0.45, 'scuba diver': 0.45, 'cougar': 0.45, 'feather boa': 0.42, 'african bush elephant': 0.4, 'siamese cat': 0.35, 'lampshade': 0.35, 'barbershop': 0.35, 'baby bib': 0.26, 'freight car': 0.11, 'laptop computer': 0.05, 'sea slug': 0.04, 'shoe store': 0.03, 'cricket insect': 0.02, 'baluster handrail': 0.0

### K.2    OPENIMAGES-100

'bee': 0.03, 'pizza': 0.08, 'goose': 0.09, 'menu': 0.23, 'lion': 0.21, 'banana': 0.14, 'umbrella': 0.21, 'jellyfish': 0.21, 'ice cream': 0.24, 'orange': 0.21, 'ant': 0.2, 'koala': 0.2, 'necklace': 0.22, 'flamingo': 0.24, 'vulture': 0.25, 'fly': 0.24, 'lemon': 0.21, 'wine bottle': 0.22, 'broccoli': 0.26, 'bison': 0.29, 'barn': 0.27, 'bald eagle': 0.3, 'hen': 0.27, 'stupa': 0.27, 'spider web': 0.24, 'pillow': 0.24, 'padlock': 0.23, 'macaw': 0.24, 'totem pole': 0.23, 'traffic light': 0.23, 'laptop computer': 0.23, 'bubble': 0.23, 'chickadee': 0.23, 'cucumber': 0.24, 'daisy': 0.24, 'warthog': 0.24, 'parking meter': 0.24, 'teapot': 0.24, 'junco': 0.22, 'spindle': 0.22, 'lionfish': 0.22, 'bagel': 0.22, 'cougar': 0.22, 'french bulldog': 0.21, 'mailbox': 0.19, 'hay': 0.19, 'stingray': 0.19, 'magpie': 0.19, 'wallaby': 0.19, 'vending machine': 0.19, 'macaque': 0.19, 'greenhouse': 0.19, 'espresso': 0.18, 'quail': 0.17, 'bottle cap': 0.17, 'grand piano': 0.16, 'acorn': 0.15, 'siamese cat': 0.14, 'guacamole': 0.13, 'gas pump': 0.13, 'mitten': 0.12, 'bell pepper': 0.12, 'fig': 0.12, 'bookstore': 0.11, 'barbershop': 0.11, 'lifeboat': 0.11, 'peafowl': 0.11, 'great white shark': 0.11, 'mongoose': 0.11, 'suspension bridge': 0.11, 'tarantula': 0.11, 'marimba': 0.11, 'dishwasher': 0.11, 'stone wall': 0.1, 'christmas stocking': 0.09, 'bolete': 0.09, 'lhasa apso': 0.09, 'soccer ball': 0.09, 'beer bottle': 0.1, 'border terrier': 0.09, 'howler monkey': 0.09, 'lampshade': 0.09, 'african bush elephant': 0.05, 'scuba diver': 0.06, 'mashed potatoes': 0.05, 'english springer spaniel': 0.04, 'cricket insect': 0.04, 'feather boa': 0.04, 'dungeness crab': 0.05, 'shoe store': 0.04, 'freight car': 0.04, 'barbell': 0.04, 'baby bib': 0.03, 'sea slug': 0.03

## L    CAPTIONNET SPREADSHEET COLUMN EXPLANATIONS

CaptionNet contains many different kinds of metadata, and the meaning of some of the column labels used may not be immediately apparent to the reader.

We do not provide explanations for metadata columns which are explained in one of the original dataset descriptions; for those, we recommend referring to the original authors of the datasets. (Deng et al., 2009; Fang et al., 2022; Schuhmann et al., 2021a; Thomee et al., 2016; Kuznetsova et al., 2020)

**BLIPCaption** refers to captions generated by us using a BLIP captioning model. **BLIPTitle** captions are a combination of the BLIP caption and the title field of flickr captions. Li et al. (2022)

**FlickrCaption** refers to captions sourced from flickr.

**annot_caption** refers to OpenImages captions that were authored by human image annotators. **prose_caption** combines BLIP and annotator captions, favoring the latter when available.

**clip_idx** are ImageNet labels chosen by a zero-shot CLIP ViT-L model from OpenAI.

**idx_** labels refer to labels generated using various subset-matching strategies.

**mc** is multiclass, **sc** is single class, **strict** is strict. **Ours, default, openai** refer to the three different sets of class labels we experimented with throughout this paper.

