# OpenReview forum: "Caption supervision enables robust learners: a controlled study of distributionally robust model training"
_ICLR.cc/2023/Conference — Submitted to ICLR 2023_

### Official Review · Reviewer_ybgD · 2022-10-25

**Confidence:** 4
**Correctness:** 3
**Technical Novelty And Significance:** 3
**Empirical Novelty And Significance:** 3
**Recommendation:** 6

**Clarity, Quality, Novelty And Reproducibility:**

The novelty and reproducibility are decent. However the clarity and writing quality can be largely improved.

**Strength And Weaknesses:**

Strength:
- Thorough study on the robustness of recent popular VL models vs conventional CE models. The findings of (2)(3)(4)(5) mentioned in the summary section above are especially interesting and can be helpful to the community when comparing new VL methods with existing methods.
- Many interesting robustness capabilities of VL models only emerge when trained with hundreds of millions of samples. With the proposed CaptionNet dataset, it is now possible to have a fairer comparison between VL models and CE models.
- Figure 1 is helpful in comprehending the effect of different loss functions vs robustness.

Weaknesses
- The writing and the paper organization can be improved. Sec 3.1 contains results on captionnet, Sec 4 also contains results on captionnet. It’s not easy to follow what the authors try to convey quickly at first glance.
- The authors perform quite a lot of ablation studies. However, there are many numbers in different tables. It is also hard to navigate through all the information due to different ablation targets. The experiments cover loss functions, label noise, caption length, caption quality, VL vs CE etc. However, the text in the current form does not lead the readers well through different sections and contents.
- There are also many typos in the text, for example page 2: “Our results that when” -> “Our results show that when”, page 5: “CaptionNet subset can be found in Section ??” -> missing reference number.
- The captions are too small to read in Figure 2 & 3.


**Summary Of The Paper:**

The authors investigate the robustness of the model trained to distribution shifts by varying the loss functions, datasets, and the selection of targeted shifts. They found that (1) models trained by Vision-Language (VL) and Cross-Entropy (CE) loss have drastically different robustness even when the training sets are in similar sizes; (2) when trained with high label noise, VL models tend to be more robust in low accuracy regime; (3) VL models enjoy longer captions; (4) When caption quality is low, longer and more descriptive captions can still benefit the training; (5) CE models can be robust but require much longer training time; (6) CE models are more sensitive to label noise than VL models. Besides the findings, the authors also present a new combined dataset called CaptionNet by sourcing 4 existing datasets and augment with over 50k samples from Flickr.

**Summary Of The Review:**

I enjoy reading the findings in this submission. However the text needs major revision.

---

> ### Author Response · Authors · 2022-11-18
> **Thank you for your feedback**
>
> We thank the reviewer for their comments and appreciate their detailed and thoughtful response. As per your recommendations, we have now clarified the abbreviations. See our main response for these and other changes we have made in this revision. We have also addressed the clarity issues by defining the abbreviations and keeping them consistent for ease of reading. We welcome additional discussion in case of any lingering issues.

---

### Official Review · Reviewer_hwfD · 2022-10-26

**Confidence:** 3
**Correctness:** 2
**Technical Novelty And Significance:** 2
**Empirical Novelty And Significance:** Not applicable
**Recommendation:** 1

**Clarity, Quality, Novelty And Reproducibility:**

Many details of created dataset and experimental setup are not discussed or are ambiguous (see Weaknesses). No code is available (yet?), and the proposed dataset is not accessible either (yet?).

**Strength And Weaknesses:**

**Strengths:**

- The proposed experimental setup is potentially very interesting as it allows to compare different training strategies relatively fairly: using image captions vs. class labels
- The authors experiment with many different datasets also using combinations of them

**Weaknesses:**

- The paper is not written clearly, creating some ambiguity and possible misunderstandings, also is not self-contained. To give some examples:
    - (Section 2, first paragraph) The authors mention using some “subset matching” technique to train models on unlabeled images. That technique, however, is not properly explained - not even in the sense of how the method itself works but also just what does this technique supposed to do. The details in the appendix only mention that the method matches samples with classes, but nothing says what this matching is based on. Only looking at Feng et al. (2022) that was referred to I can guess that the authors mean “substring matching” (not “subset”) where caption strings are matched to label names. But as this “subset matching” is an important part to understand for interpreting the results and observations, it should be really clear what this is, limiting the risk of misunderstandings.
    - (Table 2) The authors argue that “labeling strategy” is important and choosing different terms for matching has a significant impact on robustness. However, they do not explain what’s their proposed terms for matching, how they are obtained, what are they based on, etc. This makes drawing any conclusions from the results difficult.
    - (Section 3, beginning) ImageNet-100 explained as a “superset of ImageNet”. I would suppose it’s a union of samples in ImageNet-Captions and ImageNet-100 + some extra captions for 50,000 samples? The need for extra captions are motivated for balancing classes - but are they perfectly balanced then? The number of samples in Table 3 is 124k - shouldn’t that be 130k then?
    - “we use SimCLR transformations rather than CLIP transformations for all model training on CaptionNet” - what type of transformations? Are these transformations for data augmentation?
    - Are the evaluation numbers reported in Table 3 computed on the listed datasets for each row or on ImageNet-100 (the last paragraph of the experimental setup seems to mention that)? This should be clear to the reader without any ambiguities, as it’s very important for interpreting the results.
    - (Section 4.1 + Table 3) VL models are referred to as more tolerant of label noise than CE models, where OpenImages-100 is given as an example of a dataset with noisy labels. However, looking at Table 3, second row (oi100) it seems that it’s VL models that get much lower accuracy and robustness than CE models, which appears to be the opposite of what the authors claim
- Some of the conclusions and observations are not explicitly justified or informative/clear:
    - (Section 5, point 1)  The authors say “losses matter“, “can have a substantial effect on model robustness” - however, the exact effect/relation observed is not summarized here, although some of the relations are discussed throughout the paper
    - (Section 5, point 3) “Length matters” - this is something that I don’t see explicitly discussed in the paper. It should be clear which specific results indicate that

Minor:
- The links to figures and tables often don’t work/point to wrong places in the paper

**Summary Of The Paper:**

The authors construct extended datasets like ImageNet & OpenImages with additional image captions and class labels which allows a more fair comparison between training models on image captions vs. standard class labels. The authors experiment with increasing the training dataset in size but also varying different types of captions/labels with respect to their acquisition source and noise level.

**Summary Of The Review:**

The authors study a potentially interesting and insightful experimental setup, however, the paper is not written clearly to the extent that it creates a significant amount of ambiguity and risk of misunderstandings, making experimental results, observations, and conclusions difficult to interpret with confidence.

---

> ### Author Response · Authors · 2022-11-18
> **Thank you for your feedback**
>
> Thank you for your detailed feedback! Regarding your questions about the availability of the dataset and the codebase, we are happy to announce that we have released both CaptionNet itself, and the framework used to train all of our models, allowing the community to inspect and modify our methodology (for instance, to explore the effective robustness of other loss functions).
>
> Regarding your question about what our terms for matching were, we have released all of our matching terms in the framework used to train all of our models (for instance, you can find the “ours” labels at /vlhub/metadata/in1k_ours.txt). We have revised the opening of Appendix E to provide additional detail on how the labels are obtained. OpenAI described their methodologies for “labeling” ImageNet classes for zero-shot CLIP in the paper which announced the model. The “default” label set was provided by Fox, E., and Guestrin, C. (n.d.) in the Coursera Machine Learning Specialization. The “ours” label set was generated by us. Finally, as to your question about what they are based on, all of the matching terms are chosen heuristically, to the best of our knowledge. We found that the heuristic changes to term matches often had substantial effect on accuracy; however, we leave the algorithmic discovery of optimal subset matching terms to future work.
>
> We apologize for any confusion we may have introduced in Section 3 by referring to ImageNet-100 as a “superset of ImageNet”; it is a superset of 100 classes from ImageNet-Captions, and ImageNet-Captions is itself a subset of ImageNet. We have rewritten the sentence to read: “ImageNet-100 (in100): A superset of 100 ImageNet-Captions classes with over 50,000 new human-authored ground-truth labels, which also includes flickr-captions and blip-captions.”
>
> To address your question about SimCLR transformations, we have clarified using language more similar to the original Santurkar et al. reference; the passage in Section 3 now reads, “We use SimCLR augmentations (resize, crop, flip, jitter, blur, grayscale) rather than CLIP augmentations (resize and crop) for all model training on CaptionNet.”
>
> The “dataset” column in tables 2 and 3 identifies the dataset we used for the experiment in the corresponding row. We have clarified the point of confusion in section 3 by adding a heading “Evaluation on CaptionNet”, in which we state, “As is standard for this type of study, we validate on ImageNet-100 validation images (in100-val), irregardless of the choice of pretraining dataset.”
>
> Regarding your concerns re: (Section 4.1 + Table 3), we suspect that the confusion derives from a certain ambiguity as to model supervision strategy; therefore, we have added the following sentence to the caption for table 3; “CE-loss models are supervised using ground-truth labels where they are available, and subset matched labels where ground-truth labels are unavailable. VL-loss models are supervised using the best available captions for the dataset.” On OpenImages-100 (and ImageNet-100), CE-loss models trained on ground truth labels outperform VL-loss models trained on noisy captions.
>
> Finally, to address your concerns about how some of our points in section 5 are supported, we have added references to section 5 which link back to the relevant experiments.

---

### Official Review · Reviewer_kSsF · 2022-10-28

**Confidence:** 4
**Correctness:** 3
**Technical Novelty And Significance:** 3
**Empirical Novelty And Significance:** 3
**Recommendation:** 5

**Clarity, Quality, Novelty And Reproducibility:**

The paper's clarity needs to be improved while the topic is interesting. If the dataset CaptionNet is released, the paper should be reproducible.

**Strength And Weaknesses:**

Strength:
- The authors provide discussions on how this paper relates to and differs from related works.
- The authors carefully designed a set of training datasets to study the effect of different factors in vision model training, including losses, label noises, caption quality, etc.
- The authors in the end highlighted a few aspects that can guide future researchers in studying vision language models.

Weaknesses:
- The paper presentation can be improved.
    - There is one missing section link and a few grammar errors.
    - The acronyms in100 and oi100 are not been officially introduced. Similarly, there are many acronyms, for example, in Table 4, are not described.
    - The experiment setup is not very clear at first sights, like how the evaluation works.
    - The authors use CLIP-loss sometimes and VL sometimes.
    - I also feel like the text is too narrative and not very essay-y.
- Although the authors show empirically that CE and VL-loss have different behaviors, there is no theoretical analysis nor any guideline on how to use the lesson to improve the loss. Can we modify the CE loss for image classification so that it is less prone to label noises, or can we modify the VL loss so that it can learn more from clean data?
- The authors mention in 4.1 that oi100 has 90% label noise; where does this number 90% come from?
    - Also for this section, the authors say that the noises are not equal while they are both around 90% accuracy. I wonder if the authors have looked at top-5. It is possible that the clip label noises are more reasonable while the subset matching label noises are more wild.

**Summary Of The Paper:**

In this paper, the authors try to understand why CLIP-like models can have great robustness on natural distribution shifts. To understand the difference, the authors collect a dataset CaptionNet and design a careful control experiment upon it. The authors show that standard classification cross-entropy loss can also be robust in some cases. The authors did many ablation studies to show how robustness and accuracy are affected by different factors.

**Summary Of The Review:**

Overall, this paper is investigating an interesting problem and conveys a clear message in the end. However, the experiments between the motivation and conclusion are hard to follow. The paper has great potential to be impactful but it is not ready for publishing yet.
I am not sure if ICLR allows submitting a newer version of the paper. If a better manuscript is not allowed to submit, I am inclined to reject it.

---

> ### Author Response · Authors · 2022-11-18
> **Thank you for the feedback**
>
> Thank you for your feedback on our paper!
>
> Regarding your question about the 90% label noise statistic, we have rewritten section 4.2 to clarify what we meant; simply put, even the best subset matching techniques apply correct labels to less than 10 percent of the available data, suggesting that the caption text and the ground truth labels have very little overlap in OpenImages-100.
>
> We thank the reviewer for the suggestion about modifying CE-loss or VL-loss to remedy their weaknesses. We agree this would be an excellent direction for future research, and have added a comment to that effect in section 5 of the revised manuscript.
>
> Regarding your questions about the availability of the dataset and the codebase, we are happy to announce that we have released both CaptionNet itself, and the framework used to train all of our models(anonymized links in the abstract), allowing the community to inspect and modify our methodology (for instance, to explore the effective robustness of other loss functions).
>
> We appreciate your taking the time to cite technical errors and make organizational recommendations for our paper; please see our main response for a detailed list of the clarifications and formatting corrections we have added to the paper.
>
> While we agree that new theoretical results on distributional robustness would be welcome, our paper, in keeping with the vast majority of papers we cite in table 1, is a large-scale phenomenological study which relies on carefully controlled experiments to reach its conclusions.

---

### Official Review · Reviewer_y4rQ · 2022-11-03

**Confidence:** 3
**Correctness:** 2
**Technical Novelty And Significance:** 2
**Empirical Novelty And Significance:** 2
**Recommendation:** 3

**Clarity, Quality, Novelty And Reproducibility:**

Clarity: this is the major part the paper could further work on, as illustrated in the weaknesses.

Novelty: there is no proposed new methodology, and the major discovery is not clearly illustrated.

Reproducibility: the authors provide the training details in the paper, but the details of subset matching seem missing.

**Strength And Weaknesses:**

Strength:

1. The authors conduct comprehensive experiments and propose a dataset CaptionNet which is a sub-set of ImageNet and with newly added 50000 images, it is class-balanced and with caption information.

Weakness:

Overall the core contribution of the work is not clearly illustrated, and some key factors, such as subset matching are not well introduced even in the supplementary material. Also for the proposed dataset CaptionNet, it is not clear how this dataset can be further used for researchers as there are no clearly targetted research topics relating the CaptionNet.

1. The core statement of the work "caption supervision enables robust learners," seems under-illustrated.
“CNNs trained on a standard cross-entropy loss also benefit from caption supervision." This expression in the abstract is not clear to me. Does it mean by subset matching labeling on unlabeled data, CE loss could be utilized and achieves similar or higher robustness compared with CLIP? After reading the paper, I am not sure in which section this point is addressed with deep discussion. Explanations are welcomed if there are misunderstandings.

1. In this work, the authors consider comparing the standard supervised training with CLIP model from the perspective of cross-entropy loss and clip-style loss. I am confused that from this perspective, does it degrade the fundamental difference between supervised and unsupervised learning into a loss comparison? is it proper to just address this as a difference in loss function?

1. Key factors are not well introduced in the paper. For subset matching, although it is used in another paper, it is still a new method that is not widely recognized by the community. Briefly introducing how it works and how it relates to your method is a necessity. However, in the supplementary section E, it is still not clear how it works and why the following evaluation in Sec.E matters.

1. For CaptionNet, the introduction is not related to what the research topics it targets at. For example, the authors can not only bold/highlight the combination of sub-datasets but also highlight/emphasize the topic that the combination targets to explore.

1. The "label noise" (sec.4.1) and "captioning strategy" (sec.4.2),  I feel like the cause for label noise or captioning strategy is the way (subset matching) used in CaptionNet to create caption labels. However, the authors use the drawback in dataset creation as the topic this dataset is to discuss.

1. There are quite an amount of unclear abbreviations and references, which makes the paper not easy to follow. For example, "in100", ''in1000", "oi100" are not first introduced as abbreviations for each sub-dataset. In Table2. it is not introduced what "Eff. Rob. Acc. Ratio" is  , which I am afraid is also not discussed in the paper. In Table 4, all the abbreviations in the first row (evaluation metrics) and names of the evaluated models are not introduced, which makes the table really hard to read and conveys little useful information currently.

**Summary Of The Paper:**

This paper argues that using caption supervision in the way of using subset matching to label unsupervised data and use it to train cross-entropy loss would achieve a similar performance on robustness as CLIP model. Meanwhile, this paper proposes CaptionNet, a 100 classes Imagenet-based dataset with captions.

**Summary Of The Review:**

Based on the weaknesses, I think the paper could be enhanced further by better illustrating their core contribution and better introducing CaptionNet by organizing it with the targeted research topics.

---

> ### Author Response · Authors · 2022-11-18
> **Thank your for the feedback**
>
> Thank you for your feedback on our paper! We apologize for the terminological confusion we introduced in Section 2 by referring to the technique utilized by Fang et al as “subset matching”; in our revision, we refer to their technique as “substring matching”. Figure 1 in our revised manuscript illustrates more clearly how the subset matching process works.
>
> We also appreciate your taking the time to cite technical errors and make organizational recommendations for our paper; please see our main response for a detailed list of the clarifications and formatting corrections we have added to the paper.
>
> *“CNNs trained on a standard cross-entropy loss also benefit from caption supervision." This expression in the abstract is not clear to me. Does it mean by subset matching labeling on unlabeled data, CE loss could be utilized and achieves similar or higher robustness compared with CLIP?"*
>
> You are correct, that is what our experiments showed; you can find the results which support these conclusions in table 2 and figure 2 of the revised manuscript.
>
> *Does it degrade the fundamental difference between supervised and unsupervised learning into a loss comparison? is it proper to just address this as a difference in loss function?*
>
>  We agree that it would be an error to treat the fundamental difference between supervised and unsupervised learning as a loss function (we define a dataset as unsupervised w.r.t a problem when no ground-truth labels exist for that dataset for that problem). However, by design, VL-loss can utilize unsupervised data for classification, and CE-loss cannot. This is because VL-loss performs a second, implicit function which we call dataset filtration. VL-loss uses text tokens to classify images, drawing from an enormous set of implicit classes (possible token sequences). Subset matching is an explicit filtration strategy we pair with CE-loss to allow it to learn unsupervised data. We have added a heading in section 2, “labeling strategies”, which we hope clarifies this point.
>
> Incidentally, if your point was that many other potential confounding factors exist, such as model architecture, other contrastive losses, robustness interventions, etc. we mention in section 1 that these causes have already been eliminated as plausible explanations for the robustness differences by Taori et al and others. As further evidence, we provide Appendix figure 3, which compares over 1000 models on our robustness measures; you will see there that the average robustness of VL-loss models consistently exceeds all other methods, holding base accuracy constant.
>
> *The "label noise" (sec.4.1) and "captioning strategy" (sec.4.2), I feel like the cause for label noise or captioning strategy is the way (subset matching) used in CaptionNet to create caption labels. However, the authors use the drawback in dataset creation as the topic this dataset is to discuss.*
>
>  Any unsupervised labeling strategy will have errors. We certainly do not argue that subset matching is the ideal method, and in Table 10 of our appendix, we document precisely how reliable it is for ImageNet-100 and OpenImages-100 (the effectiveness of the strategy varies widely depending on the dataset). We think it is a good choice for this paper because it is simple, model-free and unbiased w.r.t. distribution of model error. We have added a heading to section 2, “Why subset matching?”, with additional detail on this.
>
> Finally, we want to address your question about the research uses of CaptionNet. CaptionNet is well suited for conducting fully controlled experiments on distributional robustness. For instance, if someone was interested in replacing InfoNCE loss with some new loss function, or if some future research group came up with a new method for filtering or condensing unsupervised datasets, they could use CaptionNet to test the effectiveness of their methods across multiple pretraining datasets simultaneously, at both low and high accuracies, and they could use the wide range of results in our paper as baselines for their improvements.

---

### Official Review · Reviewer_zEXa · 2022-11-05

**Confidence:** 3
**Correctness:** 3
**Technical Novelty And Significance:** 2
**Empirical Novelty And Significance:** 2
**Recommendation:** 5

**Clarity, Quality, Novelty And Reproducibility:**

Overall the paper is clear. I have a few suggestions:
 - It would be good to provide some examples explaining what are the supervisions for both the cross-entropy loss and caption supervisions.
 - It would be very helpful to provide the definition of “robustness” with concrete description of the metric. The authors cited Taori et.al, but it is not clear how reliable those metrics are.


**Strength And Weaknesses:**

Strength: The paper is clear overall. It studies an important question in image classification with captioning supervision.

Weakness: The novelty of the paper is limited in my view. The authors conduct some experiments to show that caption supervisions are helpful to make the model more robust, but this method has been proposed before. The dataset is interesting, but its contribution may not be enough.


**Summary Of The Paper:**

This paper shows that training a model with both cross-entropy loss and caption supervision are more robust than the model trained only on cross-entropy loss. It also releases CaptioNet, which is a 100-class subset of Imagenet with caption supervision.


**Summary Of The Review:**

Based on the limited novelty and contribution, I think it does not reach the acceptance threshold marginally. I am happy to adjust my ratings after rebuttal.

---

> ### Author Response · Authors · 2022-11-18
> **Thank you for the feedback**
>
> Thank you for your feedback on our paper! Regarding your question about whether our results are novel, we agree that previous papers, including the paper introducing CLIP itself, have shown that caption supervision in some form can contribute to model robustness; however, that is not among our key findings. In fact, in Figure 2, we demonstrate that on certain datasets, the opposite can be true; models trained with cross-entropy-loss can equal or even exceed the distributional robustness of VL-loss models. This, as far as we know, is a novel result in the literature.
>
> As to the definition of robustness we apply in our paper, we have clarified some of the references in section 1; they now specify that our paper deals with distributional robustness, which we define as robustness to naturally occurring distribution shifts; we go into some detail about the various measures we apply in section 2, under the heading “Measures of effective robustness”, and we discuss the distribution shifts we use in Appendix A.
>
> We apologize for the terminological confusion we introduced in Section 2 by referring to the technique utilized by Fang et al as “subset matching”; in our revision, we refer to their technique as “substring matching”. Figure 1 in our revised manuscript illustrates more clearly how the substring matching process works.

---

### Author Response · Authors · 2022-11-18
**Main Response**

We very much appreciate the reviewers taking the time to cite technical errors and make organizational recommendations for our paper.

Since the vast majority of issues raised by the reviewers were about clarity issues in the paper, we focus here on changes we have made to improve readability. All of our major modifications have been highlighted in blue for ease of reference.

We have added a heading to section 2, “Terminology and abbreviations”, in which we define the abbreviations used in the main paper. We have added definitions of the abbreviations used in Appendix tables 10 and 11 to the captions of those tables, since those abbreviations are not utilized in the main paper. We have added the list of shifts we used to the main body of the paper, including acronyms.

In Section 3, the CaptionNet abbreviations are now defined, and we have added a comparison table highlighting key differences between CaptionNet subsets. Effective Robustness Ratio is now always referenced using the same acronym, (E.R.R.). All Val. Acc. columns in tables in the main paper are now formatted in a uniform style, with the validation measure in parentheses: EG, Val. Acc (in100-s) for ImageNet-100-Sketch validation.

We have eliminated all references to CLIP-Loss and now we uniformly use VL-loss instead. We have fixed all missing section links. All results on CaptionNet are now contained in a single section, 4. All caption sizes are now uniform.

We hope that, taken together, these changes will help illuminate the substance of the paper and our unique contributions to the literature.

---

### Decision · Program_Chairs · 2023-01-20

**Decision:**

Reject

**Justification For Why Not Higher Score:**

No reviewer argues for acceptance (except a single "marginally above threshold").

**Justification For Why Not Lower Score:**

N/A

**Metareview: Summary, Strengths And Weaknesses:**

This paper compares the effect of captions for training vision models, through the classic cross-entropy vs a vision-language loss (e.g. CLIP). It reports a number of findings, e.g. about robustness to noise, effect of different aspects of captions (length, descriptiveness), etc. While the evaluation is extensive and includes many datasets, several of the claims are not well justified, and reviewers raise many questions about clarity.